RESEARCH COMMUNICATION

# A pathway for Parkinson's Disease LRRK2 kinase to block primary cilia and Sonic hedgehog signaling in the brain

Herschel S Dhekne[1†], Izumi Yanatori[1†], Rachel C Gomez[1], Francesca Tonelli[2], Federico Diez[2], Birgitt Schüle[3], Martin Steger[4], Dario R Alessi[2], Suzanne R Pfeffer[1]*

[1]Department of Biochemistry, Stanford University School of Medicine, Stanford, United States; [2]MRC Protein Phosphorylation and Ubiquitylation Unit, School of Life Sciences, University of Dundee, Dundee, United Kingdom; [3]Parkinson's Institute and Clinical Center, California, United States; [4]Department of Proteomics and Signal Transduction, Max-Planck-Institute of Biochemistry, Martinsried, Germany

**Abstract** Parkinson's disease-associated LRRK2 kinase phosphorylates multiple Rab GTPases, including Rab8A and Rab10. We show here that LRRK2 kinase interferes with primary cilia formation in cultured cells, human LRRK2 G2019S iPS cells and in the cortex of LRRK2 R1441C mice. Rab10 phosphorylation strengthens its intrinsic ability to block ciliogenesis by enhancing binding to RILPL1. Importantly, the ability of LRRK2 to interfere with ciliogenesis requires both Rab10 and RILPL1 proteins. Pathogenic LRRK2 influences the ability of cells to respond to cilia-dependent, Hedgehog signaling as monitored by Gli1 transcriptional activation. Moreover, cholinergic neurons in the striatum of LRRK2 R1441C mice show decreased ciliation, which will decrease their ability to sense Sonic hedgehog in a neuro-protective circuit that supports dopaminergic neurons. These data reveal a molecular pathway for regulating cilia function that likely contributes to Parkinson's disease-specific pathology.
**Editorial note:** This article has been through an editorial process in which the authors decide how to respond to the issues raised during peer review. The Reviewing Editor's assessment is that all the issues have been addressed (see decision letter).
DOI: https://doi.org/10.7554/eLife.40202.001

*For correspondence:
pfeffer@stanford.edu

[†]These authors contributed equally to this work

## Introduction

Although the cause of most Parkinson's disease remains unknown, autosomal dominant mutations in the leucine rich repeat kinase 2 (LRRK2) gene account for 1–2% of all cases (and 18% of cases in those of Ashkenazi Jewish descent) and lead to activation of the LRRK2 kinase (*Alessi and Sammler, 2018*). Recent mass spectrometry efforts have revealed a subgroup of Rab GTPases as the primary substrates of LRRK2 (*Steger et al., 2016*; *Steger et al., 2017*). Rab GTPases are master regulators of all membrane trafficking events in eukaryotic cells (*Wandinger-Ness and Zerial, 2014*; *Pfeffer, 2017*). LRRK2 phosphorylation modifies the so-called switch 2 regions of Rab proteins that report the identity of the bound nucleotide. Because cognate effector and regulatory proteins interact with Rabs in a nucleotide-dependent manner via this motif, phosphorylation on the switch two loop interferes with the ability of Rabs to bind to most of their partner proteins (*Steger et al., 2016*). Given the importance of the LRRK2 kinase in familial Parkinson's disease, there is great interest in determining which Rab-dependent processes are most significantly impacted by elevated LRRK2 kinase activity.

We recently showed that Rab29 GTPase, localized on the Golgi, recruits the predominantly cytoplasmic LRRK2 onto membranes and activates it's ability to phosphorylate downstream Rab8A and Rab10 substrates (*Purlyte et al., 2018*; see also *Liu et al., 2018*). Although Rab8A and Rab10 have distinct localizations and roles in different tissues, they (along with Rab29) have been implicated in the process of ciliogenesis. Primary cilia are single, microtubule-based cellular projections on many types of cells that are critical for developmental signaling via the Sonic hedgehog (Shh) pathway as well as for G-protein coupled receptor signaling pathways; they are produced in tight coordination with the cell cycle (*Hilgendorf et al., 2016*; *Rohatgi et al., 2007*; *Yoshimura et al., 2007*; *Nachury et al., 2007*) showed that exogenous expression of Rab8A protein increased cilia length significantly, consistent with Rab8A being a limiting component of cilia formation. *Yoshimura et al. (2007)* also discovered that the basal body associated, Cenexin-3 protein binds Rab8A and relocates to the primary cilium together with Rab8A upon serum starvation. Subsequent work has shown that both Rab8A and its guanine nucleotide exchange factor, Rabin8, are needed in part, for so-called BBSome function in ciliogenesis (*Nachury et al., 2007*; *Knödler et al., 2010*; *Westlake et al., 2011*; *Feng et al., 2012*) and also for the delivery of membrane vesicles to the growing cilium (*Lu et al., 2015*). Rab10 is detected in association with primary cilia in cultured MDCK and LLC-PK1 kidney cells and in rodent renal tubules (*Babbey et al., 2010*). Rab10 is also reported to be necessary for axonal membrane trafficking and axonal elongation (*Wang et al., 2011*; *Liu et al., 2013*).

We show here that Rab8A and Rab10 have opposing roles in normal ciliogenesis. Unlike Rab8A, Rab10 is a native suppressor of ciliogenesis, and upon LRRK2 phosphorylation, it binds preferentially to RILPL1 (Rab interacting lysosomal protein-like 1) protein to further interfere with cilia formation. These cilia modulatory events have important consequences in the brain where they influence the ability of cholinergic neurons to generate cilia in the somatosensory cortex; loss of cilia will decrease the ability of these neurons to respond to a Sonic hedgehog signal that triggers neuroprotective signaling toward dopaminergic circuits (*Gonzalez-Reyes et al., 2012*).

## Results

### Rab8A increases while Rab10 inhibits cilia formation in A549 cells

To explore the relationship between primary cilia and two important LRRK2 substrates, we revisited the contributions of Rab8A and Rab10 to primary cilia formation. Throughout the text, Rab8 will refer to Rab8A unless otherwise indicated. Consistent with previous reports (*Yoshimura et al., 2007*; *Nachury et al., 2007*; *Feng et al., 2012*; *Knödler et al., 2010*; *Sato et al., 2014*; *Babbey et al., 2010*), some of the total pool of Rab8A localized at the base and within primary cilia (detected with anti-Arl13b antibody; *Figure 1A*). In contrast, a portion of the total pool of Rab10 localized only to the base of cilia in both hTERT-RPE as well as primary, antibody-panned, rat astrocyte cells (*Figure 1A*). To study the roles of Rab8A and Rab10 in primary cilia formation, the proteins were knocked out of A549 cells using CRISPR-Cas9, and their depletions were confirmed by immunoblot (*Figure 1—figure supplement 1*). As shown previously, loss of Rab8A decreased significantly the capacity of A549 or RPE cells to generate primary cilia (*Figure 1B–D*). Surprisingly, loss of Rab10 increased the percentage of cells that made primary cilia compared with parental wild type A549 and RPE cells (*Figure 1B,C,E*). Importantly, the effects of Rab8A or Rab10 knockout on primary cilia formation were reversed upon re-expression of exogenous Rab8A or Rab10 wild type proteins (*Figure 1D,E*; Rab10,~4 fold overexpressed and Rab8A,~10 fold): exogenous Rab8A stimulated while Rab10 inhibited ciliogenesis in cells lacking these proteins. Note that RPE cells start at a higher level of intrinsic ciliation (*Figure 1C*, right) and thus there is less possibility of enhancing ciliation in this cell type.

Finally, to investigate the relationship between Rab8A ciliogenesis activation and Rab10 suppression, A549 cells lacking Rab8A were further depleted of Rab10 using a lentiviral shRNA. More than 90% of the endogenous Rab10 was depleted in these cells as seen by immunoblot (*Figure 1—figure supplement 1*, right panel). Surprisingly, loss of both Rab8A and Rab10 from A549 cells reversed the block of ciliogenesis observed in Rab8A KO cells (*Figure 1F*). Thus, Rab10 is a dominant suppressor of cilia formation. Even though A549 Rab10 KO cells are significantly ciliated, exogenous Rab8A even further increases ciliation in these cells (*Figure 1G*). Thus, Rab8A drives a Rab10-independent pathway for cilia formation. Note that loss of either Rab10 or Rab8A does not change the

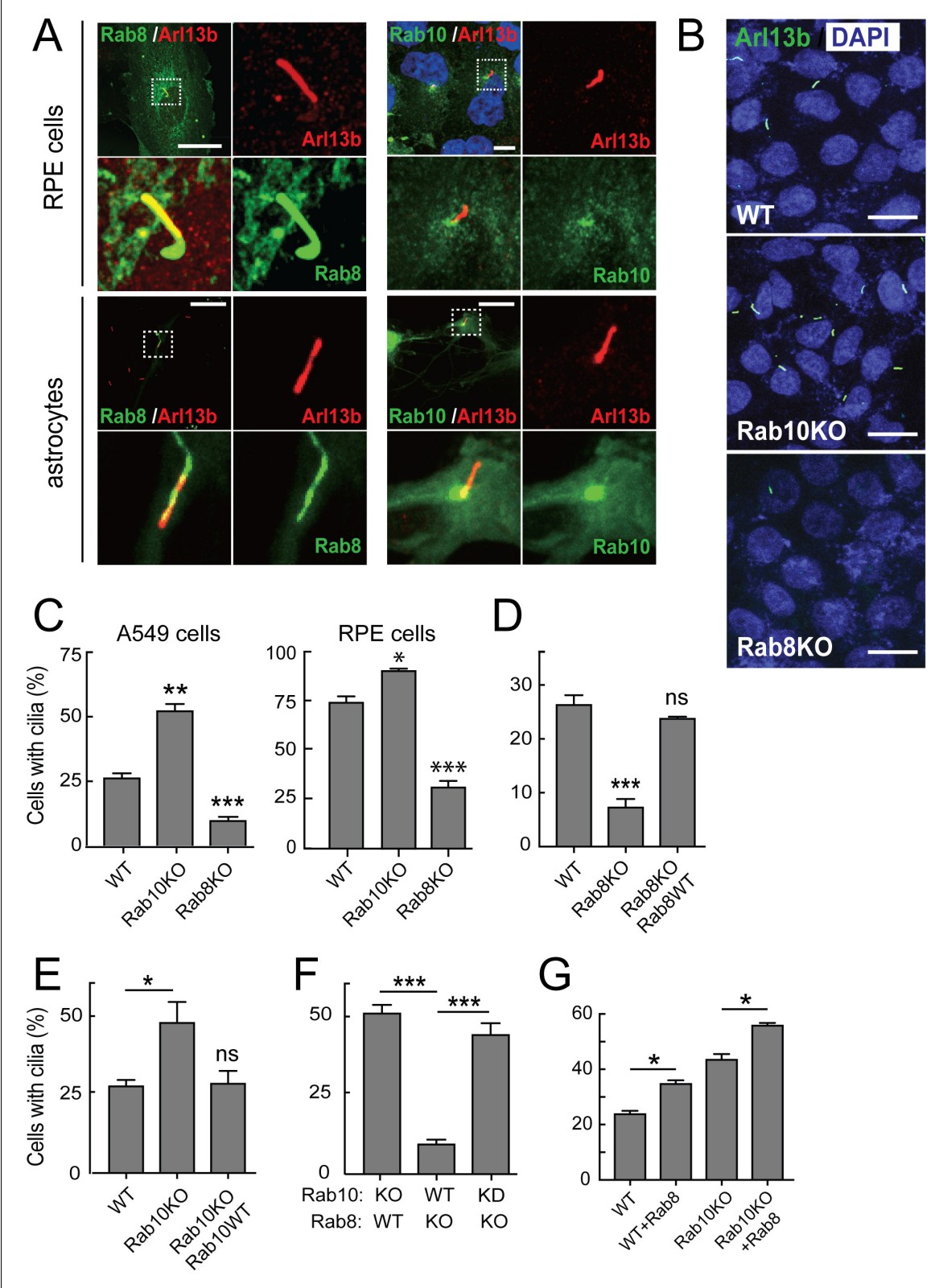

**Figure 1.** Rab10 is a negative regulator of primary cilia formation. hTERT-RPE cells (*Figure 1A*, upper panels) or primary rat astrocytes (*Figure 1A*, lower panels) were transfected with GFP-Rab8A (A, left) or GFP-Rab10 (A, right). 24 hr post transfection, cells were fixed and stained for primary cilia using anti-Arl13b antibodies (red). Dotted boxes indicate areas that are magnified in the insets. Yellow indicates co-localization between GFP-Rabs (green) and Arl13b (red). (B) A549 knock out cells lacking endogenous Rab10 (Rab10-KO) or Rab8A (Rab8-KO) were ciliated on glass coverslips and 48

*Figure 1 continued on next page*

*Figure 1 continued*

hr later fixed and stained for primary cilia Arl13b (green). (C) Quantitation of data shown in (B) for A549 cells and also RPE cells analyzed in parallel. Error bars represent SEM from three experiments with >200 cells per condition in each experiment. (D,E) Rab8A-KO or Rab10-KO A549 cells were infected with lentivirus to stably express wild type GFP-Rab8A or GFP-Rab10, respectively. Cells were ciliated on coverslips and stained for Arl13b. (D) Quantitation of primary cilia in Rab8A-KO A549 cells stably expressing GFP-Rab8A-WT. (E) Quantitation of primary cilia in Rab10-KO A549 cells stably expressing GFP-Rab10-WT. (F) Quantitation of primary cilia in Rab8A-KO A549 cells treated with Rab10 shRNA. (G) Quantitation of primary cilia in WT and Rab10-KO A549 cells expressing GFP-Rab8A. Error bars represent SEM from three experiments with >200 cells per condition. *, p<0.05; **, p<0.01; ***. p<0.001; ns = not significant. Scale bars = 10 µm.

DOI: https://doi.org/10.7554/eLife.40202.002

The following figure supplement is available for figure 1:

**Figure supplement 1.** Documentation of Rab8A and Rab10 shRNA efficacy.

DOI: https://doi.org/10.7554/eLife.40202.003

level of the other protein (*Figure 1—figure supplement 1*). In addition, Rab8 goes to the cilium in Rab10KO cells, as in wild type cells.

## Rab8A and Rab10 mutants cannot be used to mimic phosphorylation

Phosphorylation of Rab proteins by LRRK2 has been shown to interfere with their ability of to bind to numerous, cognate effector proteins (*Steger et al., 2016*). Studies of protein phosphorylation often utilize substrate mutants generated in the hope that they cannot be phosphorylated (T-to-A) or alternatively, that they will mimic the phosphorylation state (T-to-E). Unfortunately, we were unable to use such mutant Rab8A or Rab10 proteins as a series of functional and localization tests showed that they are nonfunctional. The non-phosphorylatable, threonine to alanine (TA) mutations in both Rab8A and Rab10 failed to reverse their respective ciliation phenotypes observed in knock-out cells (*Table 1*), indicating that they are non-functional. Light microscopy of A549 cells expressing these mutants showed that unlike wild type Rab8A protein, Rab8A-TA was more concentrated over the Golgi and not seen in cilia; Rab8A-TE was more dispersed than the wild type protein and may aggregate upon expression in HEK293T cells. Rab10-TE was much more concentrated on the Golgi than the wild type protein and Rab10-TA was entirely cytosolic (*Figure 2A*), as confirmed by membrane fractionation after expression in HEK293T cells (*Figure 2B*). In vitro experiments using biotin-geranyl pyrophosphate as substrate confirmed that neither Rab8A-TE nor Rab10-TA mutants are good substrates for Rab geranylgeranyl transferase (*Figure 2C*). Finally, unlike the phosphorylated versions of Rab8A or Rab10 proteins (*Steger et al., 2017*), Rab8A-TE and Rab10-TE failed to show enhanced interaction with GFP-RILPL1 protein (*Figure 2—figure supplement 1*). This is a very important caution—the TA and TE mutants do not reflect the desired states and should not be used in future experiments exploring the consequences of Rab8A and Rab10 phosphorylation.

**Table 1.** Summary of Rab8A and Rab10 phosphosite mutant protein properties

| Rab protein | Membrane associated | Localization | In vitro prenylated | In/at base of cilia | Rescue ciliation in knockout cells | Bind RILPL1 strongly |
|---|---|---|---|---|---|---|
| Rab8A WT | YES | Perinuclear Cilium | YES | In/at base | YES | When phosphorylated |
| Rab8A TA | YES | Golgi | YES | NO | NO | NO |
| Rab8A TE | Aggregated? | Golgi? | NO | NO | NO | POORLY |
| Rab10 WT | YES | Perinuclear Cilium base | YES | At base | YES | When phosphorylated |
| Rab10 TA | NO | Cytosol | NO | NO | NO | NO |
| Rab10 TE | YES | Golgi | YES | NO | NO (dominant effect) | POORLY |

DOI: https://doi.org/10.7554/eLife.40202.006

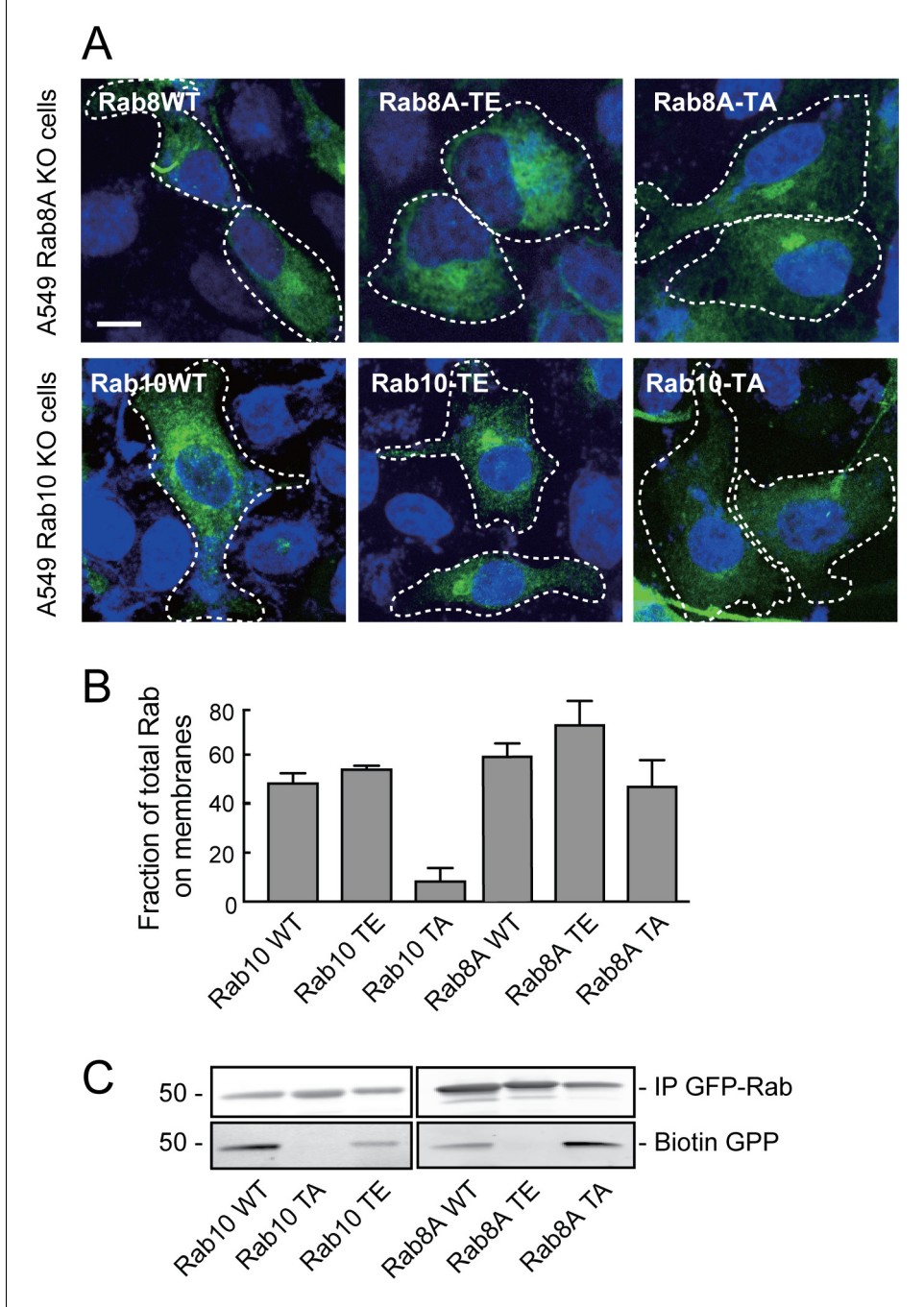

**Figure 2.** Rab8A and Rab10 phospho-site mutants are non-functional proteins. (**A**) Rab8A-KO and Rab10-KO A549 cells were infected with lentivirus to stably express wild type (WT), phospho-mimetic (TE) or non-phosphorylatable (TA) mutants of Rab8A and Rab10, respectively. 24 hr post transfection, cells were fixed and stained with anti-GFP. (**B**) HEK293T cells were transfected with wild type (WT), T73E (TE) or T73A (TA) mutant GFP-Rab10 or GFP-Rab8A as indicated. After 24 hr, cells were fractionated into membrane and cytosol and 50 μg membranes and the equivalent volume of cytosol was immunoblotted using anti-GFP antibodies. Graph indicates the percent of total Rab10 or Rab8A protein that was membrane associated. (**C**) In vitro prenylation measured using cytosol fractions from (**B**), incubated for 4 hr with biotin-geranyl pyrophosphate at room temperature. The reaction was immunoprecipitated with GFP binding protein-Sepharose and samples were immunoblotted for biotin using Strepatavidin-Alexa 800 and anti-GFP.

DOI: https://doi.org/10.7554/eLife.40202.004

The following figure supplement is available for figure 2:

*Figure 2 continued on next page*

*Figure 2 continued*

**Figure supplement 1.** Phospho-site mutants of Rab8A and Rab10 bind RILPL1 poorly.
DOI: https://doi.org/10.7554/eLife.40202.005

## Phosphorylated Rab10 clusters near peri-centriolar membranes

Rab10 is phosphorylated in cells transfected with LRRK2 kinase (*Steger et al., 2016*; *Purlyte et al., 2018*). To identify the localization of both total and phospho-Rab10 (pRab10) by immuno-staining, HeLa cells were co-transfected with the activated, pathogenic mutant LRRK2-R1441G and GFP-Rab10. After 24 hr,~80% of cells expressing exogenous LRRK2 showed total Rab10 concentrated over a perinuclear cluster (*Figure 3A,B*). Furthermore, labeling of LRRK2 expressing cells using a commercial anti-phosphoT73 Rab10-specific antibody revealed pRab10 staining only in LRRK2 co-expressing cells; the strongest pRab10 signal was also detected in the perinuclear region together with total Rab10 staining (*Figure 3C*). pRab10 localized adjacent to centrioles labeled with Centrin-3, and co-localized with perinuclear transferrin receptor (TfR)-containing membranes (*Figure 3C*). As noted earlier, RILPL1 interacts preferentially with the phosphorylated forms of Rab8A and Rab10 (*Steger et al., 2017*); *Figure 2—figure supplement 1*). In HEK293T cells co-transfected with RILPL1-GFP and LRRK2, the clustered pRab10 co-localized perfectly with RILPL1 in the peri-centriolar region. This strong co-localization supports the conclusion that RILPL1 binds to pRab10 in cells.

## RILPL1 regulates pRab10 localization

RILPL1 has been shown to associate with the mother centriole that becomes the basal body that nucleates the primary cilium (*Schaub and Stearns, 2013*); RILPL1's C-terminal half is needed for this localization. We confirmed mother centriole localization of RILPL1 in A549 cells (*Figure 3—figure supplement 1*); RILPL1-GFP co-localized with CEP164, a marker of the mother centriole. Previous work also indicated that RILPL1 influences the removal of signaling receptors from primary cilia (*Schaub and Stearns, 2013*). Primary cilia form in G1 of the cell cycle and disassemble before mitosis. In RPE cells that had been induced to form cilia by serum starvation, endogenous RILPL1 co-localized with Rab10 at the base of primary cilia, identified by staining with anti-Arl13B antibodies (*Figure 4A*). A similar localization was also seen in ciliated, primary rat astrocytes (*Figure 4B*).

When RILPL1 expression was decreased in HEK293T cells using a lentiviral shRNA, perinuclear clustering of pRab10 decreased and that phenotype was corrected upon RILPL1-GFP expression (*Figure 4D*). Indeed, perinuclear pRab10 clustering increased upon exogenous RILPL1 expression (*Figure 4D*). Overexpression of the C-terminal half of RILPL1 (RILPL1-C-GFP) dispersed pRab10 localization, whereas expression of the N-terminal half (RILPL1-N-GFP) had the same effect as the full length protein (*Figure 4C,E*). Previous work showed that overexpression of the RILPL1 C-terminal half interferes with RILPL1 localization and this region localizes RILPL1 to the centriole and dominantly blocks primary cilia formation (*Schaub and Stearns, 2013*). Altogether, these data suggest that the RILPL1 N-terminus interacts with pRab10 in pericentriolar membranes; overexpression of the C-terminus dominantly interferes with cilia formation, perhaps by competing for interactions that localize pRab10 and RILPL1 to that cellular compartment. Nevertheless, even in the absence of RILPL1, some pRab10 localized to the pericentriolar region. This may be due to association with the related RILPL2 protein still present in these cells and/or to other proteins.

Similar results were obtained for pRab10 localization in A549 RILPL1 KO cells (created using CRISPR-Cas9; protein loss was confirmed by immunoblot; *Figure 4H*). While LRRK2 transfected, parental A549 cells displayed most pRab10 in a peri-nuclear cluster, RILPL1 KO A549 cells had a less focused distribution of pRab10 and showed additional punctate structures in the cytosol (*Figure 4F–H*). These data support the conclusion that RILPL1 enhances pRab10 clustering on pericentriolar membranes.

## RILPL1 suppresses cilia formation at the mother centriole

Knockout of RILPL1 in A549 cells increased the percentage of cells that contained primary cilia, and the length of the cilia also increased (*Figure 5A,B*). This effect was abrogated upon expression of exogenous RILPL1 (*Figure 5C*), indicating that exogenous RILPL1 suppresses primary cilia formation, in wild type cells and in cells lacking RILPL1 protein. A similar inhibition of ciliogenesis was also

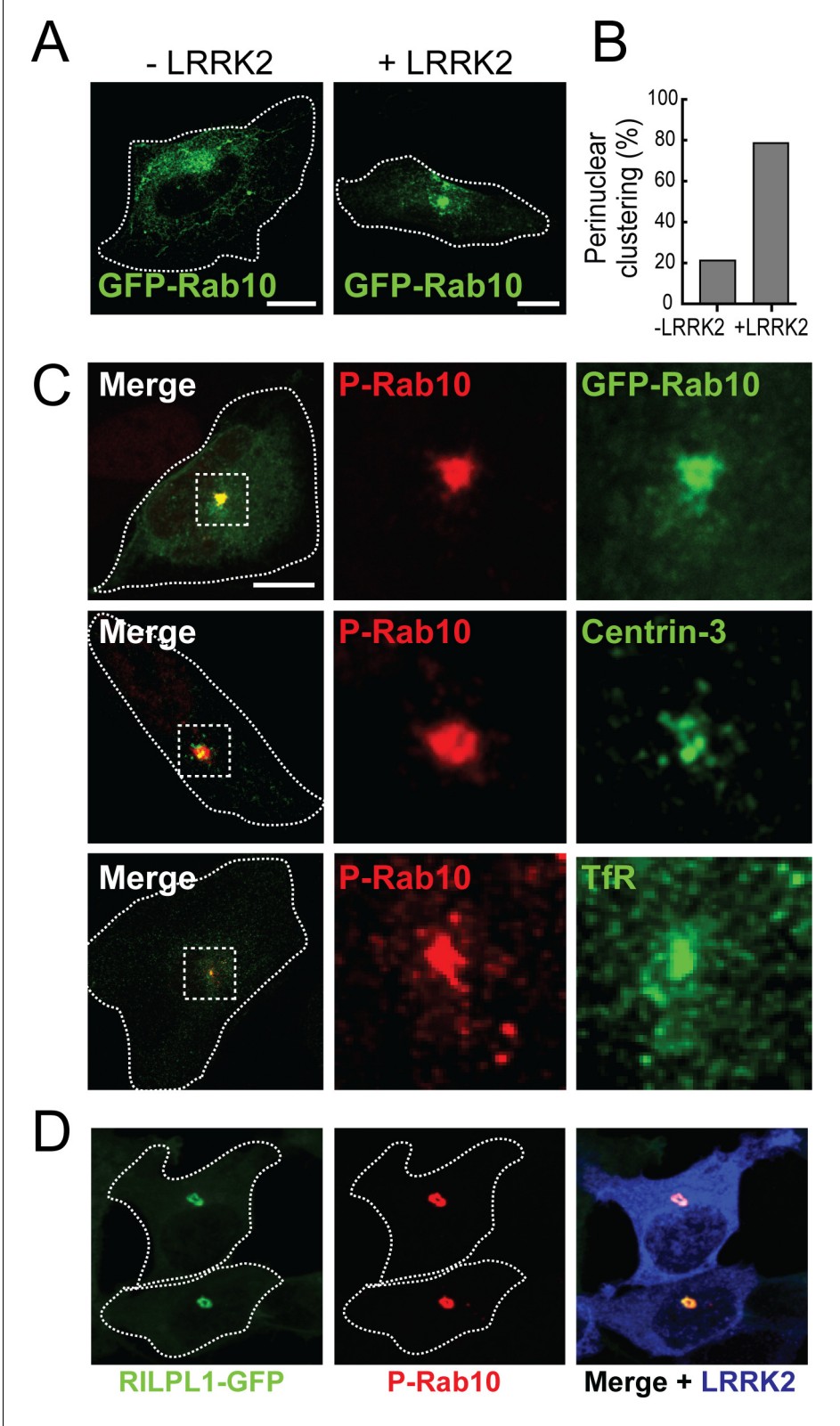

**Figure 3.** pRab10 localizes on peri-centriolar membranes. HeLa cells stably expressing GFP-Rab10 were transfected with or without LRRK2-R1441G plasmid. After 24 hr, cells were fixed and stained with the indicated antibodies. (**A**) Localization of total GFP-Rab10 (green) without (left) or with (right) R1441G-LRRK2 transfection. (**B**) Quantitation of peri-nuclear clustering of GFP-Rab10. (**C**) Cells were stained with rabbit anti-pRab10 antibody (red)

*Figure 3 continued on next page*

*Figure 3 continued*

and co-stained with either mouse anti-GFP (green), mouse anti-Centrin-3 (green) or mouse anti-transferrin receptor (TfR, green). (D) HEK293T cells were co-transfected with myc-LRRK2-R1441G and RILPL1-GFP (green). After 24 hr, cells were fixed and stained for pRab10 (red) and LRRK2 (blue). Dotted lines indicate cell boundaries. Dotted boxes indicate the enlarged region of interest. Scale bars, 10 μm.

DOI: https://doi.org/10.7554/eLife.40202.007

The following figure supplement is available for figure 3:

**Figure supplement 1.** RILPL1 localizes to mother centrioles.

DOI: https://doi.org/10.7554/eLife.40202.008

---

detected in RPE cells (*Figure 5D*). Quantitation of ciliation as a function of RILPL1 expression in RPE cells showed that the phenotype was strongest in cells showing the highest RILPL1 expression levels (*Figure 5E*).

To further study how RILPL1, Rab10 and/or RILPL1/pRab10 complexes suppress ciliogenesis, GFP-Rab10 or GFP-Rab8A were expressed in A549 cells lacking RILPL1. In this background, exogenous Rab10 inhibited and Rab8A stimulated ciliogenesis, even in the absence of RILPL1 (*Figure 5F*). This suggests that Rab8A and Rab10 do not absolutely require RILPL1 for their effects on ciliogenesis or can mediate their effects via another protein such as the related RILPL2. In contrast, when exogenous RILPL1 was introduced into cells lacking Rab8A or Rab10, it was no longer capable of repressing ciliogenesis (*Figure 5G*). These experiments demonstrate that RILPL1 needs Rab10 to repress ciliogenesis, however, exogenous Rab10 can also suppress ciliogenesis independent of RILPL1 protein (*Figure 5D,E*). Because cells lacking Rab8A are already poorly ciliated, we cannot discern further RILPL1-inhibition in this cell background.

It was recently reported that LRRK2 overexpression in HEK293T cells decreases centrosomal cohesion (*Madero-Pérez et al., 2018*). Since RILPL1 and pRab10 accumulate in the pericentriolar region, we examined the effect of exogenous RILPL1 on centriolar markers in A549 cells (*Figure 5—figure supplement 1*). As noted earlier, exogenous RILPL1 was cytosolic but also enriched at the mother centriole (*Figure 3—figure supplement 1*). Its expression was sufficient to trigger an increase in the distance between mother and daughter centrioles in some cells: specifically, 20% of A549 cells showed centrioles > 2 μm apart, whereas 48% of cells showed this phenotype upon RILPL1 expression. However, there was no relationship between RILPL1 expression level and centriolar distance over 3 logs of expression measured by quantitative light microscopy (*Figure 5—figure supplement 1*). This indicates that centriolar separation may be a secondary phenomenon, indirectly triggered in some cells by RILPL1 expression. No change in centriolar distance was detected in RILPL1 KO cells; finally, the ability of RILPL1 to increase centriole distance was completely nullified in Rab10 or Rab8A-KO cells while the RILPL1 localization remained unchanged. In summary, RILPL1 blocks ciliation by a Rab10-dependent process and may disrupts centriolar pairs in some cells under the same conditions.

## LRRK2 kinase acts via a pRab10-RILPL1 pathway to inhibit ciliogenesis

We previously reported that mouse embryonic fibroblasts (MEFs) derived from LRRK2-R1441G mice showed fewer primary cilia, a phenotype that was reversed by adding the LRRK2-specific inhibitor, MLi2 (*Steger et al., 2017*). Endogenous pRab10 was detected (for the first time) in R1441G LRRK2 MEF cells localized at the base of primary cilia (*Figure 6A*). The pRab10 staining was lost on treatment with MLi2 along with a concomitant increase in ciliogenesis (*Figure 6A,C*). Thus, LRRK2 phosphorylated Rab10 is seen concentrated at the base of primary cilia.

To investigate whether pRab10 and RILPL1 contribute to the defect in ciliogenesis observed in R1441G MEF cells, both proteins were depleted using lentiviral shRNAs. qPCR analysis showed more than 85% reduction in the levels of both mRNAs upon shRNA expression (*Figure 6B*). In control cells, RILPL1 and pRab10 localized to the base of cilia and this staining was lost upon expression of either of the corresponding shRNAs (*Figure 6A*). Importantly, when either Rab10 or RILPL1 was depleted, the effect of LRRK2 activity on cilia formation was completely abolished. These data demonstrate that the ability of pathogenic LRRK2-R1441G to influence cilia formation depends entirely on the presence of Rab10 and RILPL1 proteins.

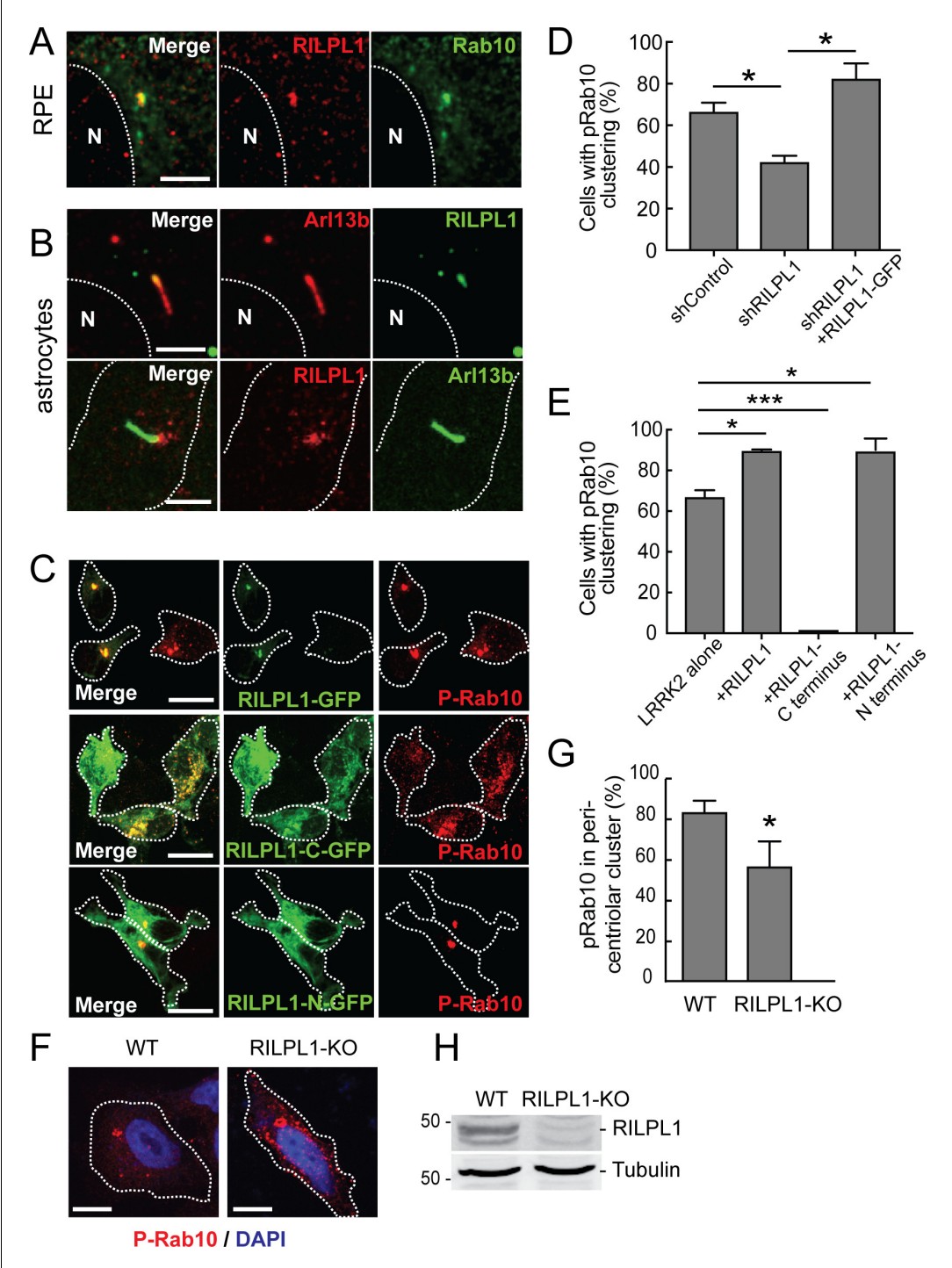

**Figure 4.** Localization of pRab10 is influenced by RILPL1. (**A**) RPE cells were ciliated for 24 hr and stained with rabbit anti-RILPL1 antibodies (red) and co-stained with mouse anti-Arl13b or rabbit anti-Rab10 antibodies (green). (**B**) Rat primary astrocytes were ciliated and stained with anti-RILPL1 and anti-Arl13B antibodies as indicated. (**C–E**) HEK293T cells were infected with lentiviral shRNA for RILPL1 and selected by puromycin. LRRK2-R1441G was co-transfected with either RILPL1-WT. Also, RILPL1-C-terminus GFP or RILPL1-N-terminus GFP (green) were co-transfected with LRRK2 in wildtype cells. After 24 hr, cells were stained with anti-phospho-Rab10 antibody (red). (**D**) Quantitation of cells with peri-centriolar clustering of pRab10 in RILPL1 knock down HEK293T cells, with or without rescue by (full length) RILPL1-WT. (**E**) Quantitation of peri-centriolar clustering of pRab10 in cells transfected with full length, RILPL1-N terminus, RILPL1-C terminus. (**F**) A549 WT and A549-RILPL1 knock out cells were transfected with myc-LRRK2-R1441G and 24 hr later, stained with anti-pRab10 (red) and DAPI (blue). (**G**) Quantitation of total pRab10 in peri-centriolar clusters. (**H**) A549 WT and A549-RILPL1 knock

*Figure 4 continued on next page*

*Figure 4 continued*

out cells were lysed and analyzed by immunoblotting with the indicated antibodies. Error bars represent SEM from three experiments with >50 cells per condition in each experiment. *, p<0.05; **, p<0.05; ***, p<0.001. Scale bar, 10 μm.

DOI: https://doi.org/10.7554/eLife.40202.009

*Figure 6D* compares two scenarios by which Rab10 and RILPL1 interfere with cilia formation. At endogenous levels of expression, we propose that pathogenic LRRK2 generates an inhibitory complex that contains pRab10 and RILPL1. RILPL1 shows enhanced binding to p-Rab10, and our data show that LRRK2 inhibition requires both of these proteins to yield a ciliation defect. In addition, in the absence of pathogenic LRRK, upon overexpression of RILPL1, we also detect inhibition of cilia formation; this inhibition requires the presence of Rab10, and may reflect a similar inhibitor complex that requires overexpression to generate in the absence of Rab10 phosphorylation. In addition, over-expressed Rab10 can also inhibit cilia formation in a RILPL1-independent manner that may involve RILPL2 or other proteins; RILPL1 appears a more potent inhibitor than Rab10 upon overexpression. Further work is needed to explore this possibility. In summary, our data support a model in which pathogenic LRRK2 alters cilia formation by phosphorylating Rab10, which recruits RILPL1 to coordinately interfere with cilia formation (*Figure 6D*, right). In the absence of LRRK2 activation, exogenous RILPL1 can also block ciliation from its location adjacent to the mother centriole; in this scenario, RILPL1 requires Rab10 protein (*Figure 6D*, left). We speculate that additional copies of exogenous RILPL1 are sufficient to bind non-phosphorylated Rab10 protein to generate a dominant inhibitor of ciliation.

## Pathogenic LRRK2 decreases sonic hedgehog signaling in MEFs and human iPS cells

How might cilia dysfunction influence Parkinson's disease pathology? Efficient Sonic hedgehog (Shh) signaling requires the ability of many cell types to form primary cilia and traffic signaling proteins such as Smoothened into the primary cilium (*Rohatgi et al., 2007*; *Caspary et al., 2007*). Shh signaling triggers a downstream pathway that increases the expression of the Gli1 transcription factor (*Dahmane et al., 1997*; *Niewiadomski and Rohatgi, 2015*). To test whether LRRK2 interference with ciliogenesis alters the ability of cells to respond to Shh, R1441G-expressing MEF cells were treated with Shh, with or without the LRRK2 inhibitor, MLi2; the amount of Gli1 transcript produced was then monitored by qPCR. As shown in *Figure 7A*, Gli1 expression increased upon addition of Shh; moreover, inhibition of the LRRK2 kinase in these cells resulted in an almost two fold increase in the response of these cells to Shh. These experiments show clearly that the ability of LRRK2 to block cilia formation correlates directly with the ability of LRRK2 expressing cells to carry out Shh signal transduction.

Similar results were obtained using patient-derived, induced pluripotent stem cells (iPS cells) carrying the LRRK2 G2019S mutation (*Figure 7B–E*). Undifferentiated LRRK2 G2019S iPS cells (LRRK2[G2019S/WT]), and the corresponding, zinc finger-corrected wild type cell line (LRRK2[WT/WT]; *Sanders et al., 2014*) were plated on matrigel-coated coverslips and stained for the presence of cilia after 72 hr. Cells carrying the G2019S mutation showed 50% less ciliation than the corresponding, corrected iPS cells; the length of the cilia were comparable (*Figure 7B,D*). Importantly, the extent of ciliation in the G2019S iPS cells was fully restored upon treatment with MLi2 (*Figure 7C*). Like their R1441G LRRK2 MEF cell counterparts, LRRK2[G2019S/WT] iPS cells showed 50% less Gli1 expression than wild type cells, and this reduction was directly proportional to the extent of their loss of cilia (*Figure 7C,E*). These experiments demonstrate that pathogenic LRRK2 proteins can influence Shh signaling by changing the extent of ciliation in multiple cell types. That a heterozygous LRRK2 mutant iPS cell line showed this defect was striking.

## LRRK2 R1441C mice show cilia defects in the brain

To study further whether primary cilia defects seen in cell culture are also present in the brain, brain sections from 7 month old, LRRK2 R1441C mutant mice were compared with age- and litter-matched wild type animals. Sagittally sectioned brain slices were stained with neuron-specific primary cilia markers: adenylate cyclase 3 (AC3), the major neuron-specific primary cilia marker, and somatostatin

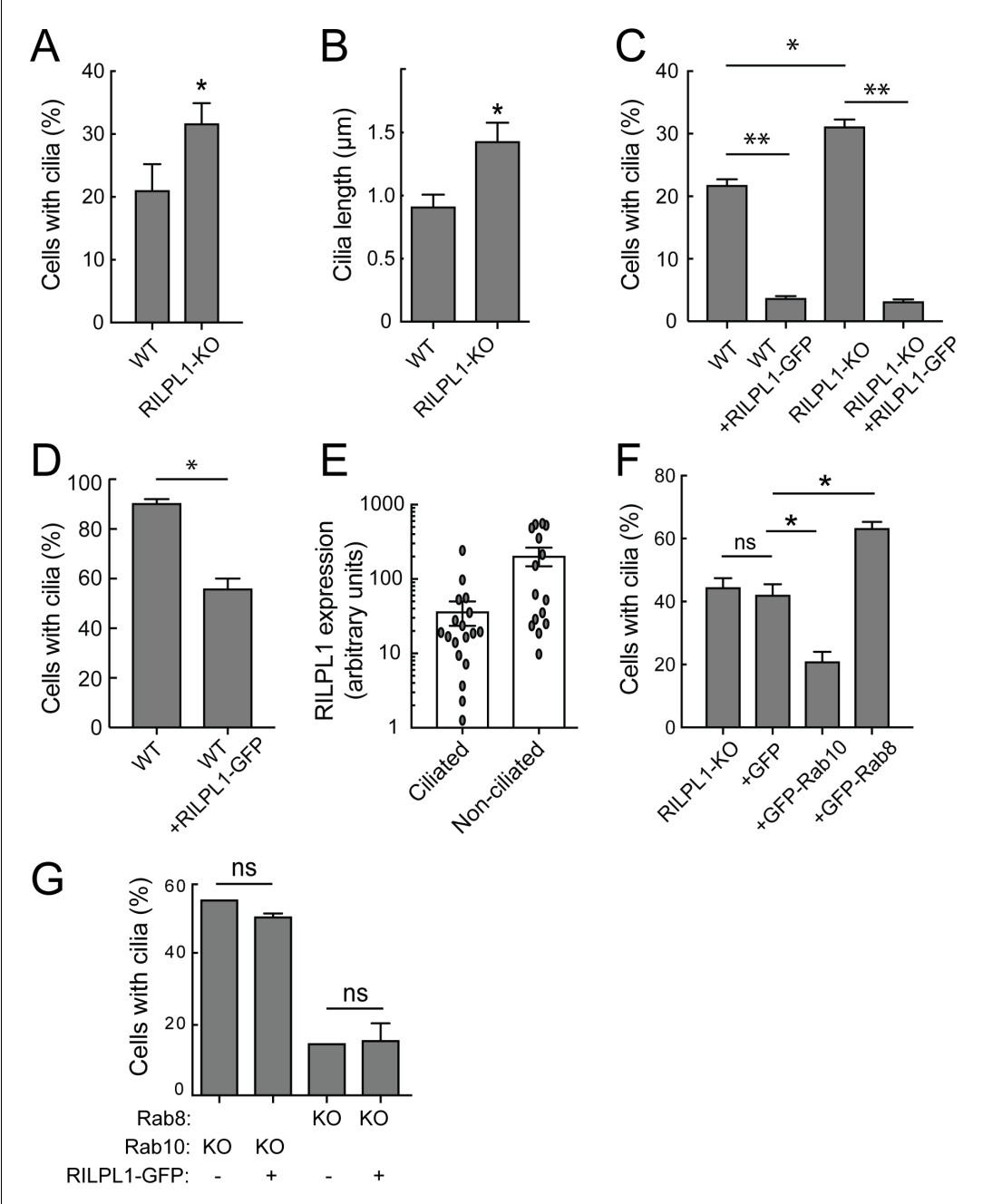

**Figure 5.** RILPL1 is a negative regulator of ciliation. (**A**) Quantitation of primary cilia in WT and RILPL1-KO A549 cells. (**B**) Quantitation of cilia length in WT and RILPL1-KO cells. (**C**) Quantitation of cells with cilia when RILPL1-GFP is expressed in WT or RILP1-KO ad A549 cells. (**D**) Quantitation of cells with cilia when RILPL1-GFP is expressed in WT hTERT-RPE cells. (**E**) Quantitation of RILPL1-GFP expression in ciliated versus non-ciliated cells RPE by determining GFP signal intensity per cell in microscopy images. (**F**) Quantitation of cilia in A549 RILPL1-KO cells that were either un-transfected or transfected with GFP alone, GFP-Rab10 or GFP-Rab8A. (**G**) Quantitation of cilia in A549 cells lacking Rab8A or Rab10 as indicated, with or without RILPL1-GFP expression.

DOI: https://doi.org/10.7554/eLife.40202.010

The following figure supplement is available for figure 5:

**Figure supplement 1.** Exogenous RILPL1 expression disrupts centriolar cohesion.

DOI: https://doi.org/10.7554/eLife.40202.011

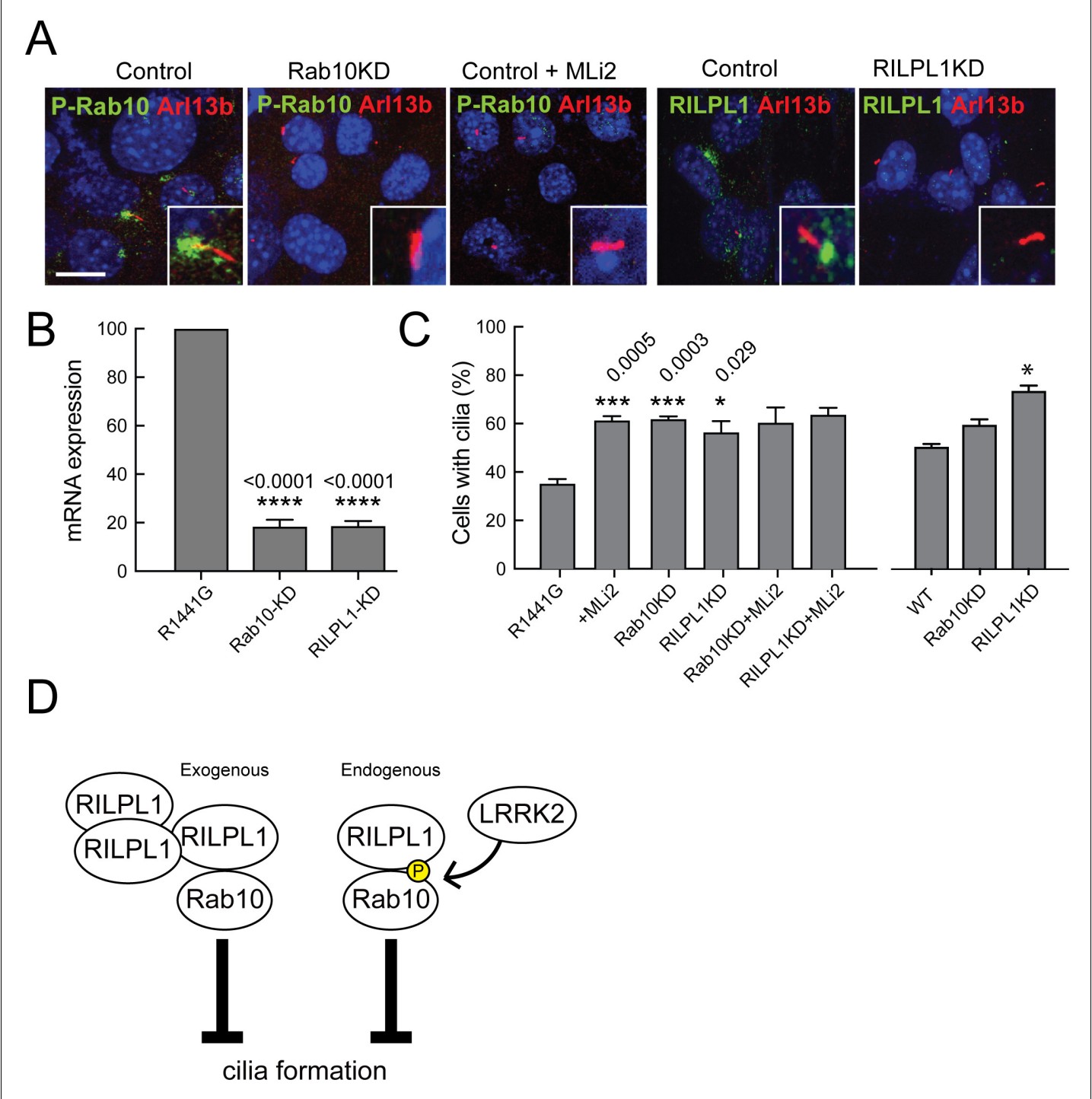

**Figure 6.** LRRK2-mediated cilia defects require Rab10 and RILPL1. Rab10 and RILPL1 were depleted from LRRK2 R1441G MEFs using lentiviral delivery of shRNA. Cells were plated on coverslips or lysed for qPCR analysis 4 days post-infection. After 48 additional hours, cells were serum starved for 16 hr in the presence of 200 nM MLi2 or DMSO as indicated. (**A**) Cilia were stained with anti-Arl13b antibodies (red) and either anti-pRab10 (green) or anti-RILPL1 (green) as indicated. Nuclei were stained with DAPI (blue). Scale bar, 10 µm. Inset boxes show magnified regions including the cilium and its base. (**B**) Relative mRNA expression for Rab10 and RILPL1 in Rab10KD and RILPL1KD normalized to control R1441G-MEF cells. Error bars indicate standard error of the mean from quadruple qPCR replicates. (**C**) Quantitation of primary cilia in (left) R1441G-MEF cells or (right-most three bars) WT MEF cells,±MLi2,±Rab10 knock down or ±RILPL1 knock down. Error bars represent SEM from duplicate experiments quantifying >100 cells per experiment. Student unpaired 2-tailed T-test values showing *,<0.05, **,<0.001. (**D**) Graphical representation of the working model in which RILPL1 and phosphorylated Rab10 suppressing cilia formation cooperatively. RILPL1 cannot inhibit ciliation in cells lacking Rab10; overexpression of Rab10 inhibits

*Figure 6 continued on next page*

*Figure 6 continued*
ciliation, and LKKR2 kinase increases the susceptibility of cells to RILPL1 inhibition by increasing the affinity of RILPL1 for p-Rab10 protein. Under these conditions, pRab8A can also bind RILPL1 and both Rab8A and Rab10 fail to bind most of their other effector proteins.
DOI: https://doi.org/10.7554/eLife.40202.012

receptor 3 (SSTR3) that is also found in primary cilia in certain brain regions (*Händel et al., 1999*; *Bishop et al., 2007*; *Berbari et al., 2008*). According to the human protein atlas and (*Mandemakers et al., 2012*; *West et al., 2014*), LRRK2 expression is highest in the cerebellum, hippocampus and cortex (*Figure 8G*). The majority of the neurons in the cortex, striatum and hippocampus (but not the cerebellum) had AC3$^+$ primary cilia. SSTR3$^+$ cilia were found in the cortex and the hippocampus but not in the striatum, presumably because striatal neurons themselves secrete somatostatin to signal to other cells.

Primary cilia length and density varied between these brain regions. Consistent with data from MEF-R1441G and human iPS cells, fewer neurons in the somatosensory cortex of R1441C mouse

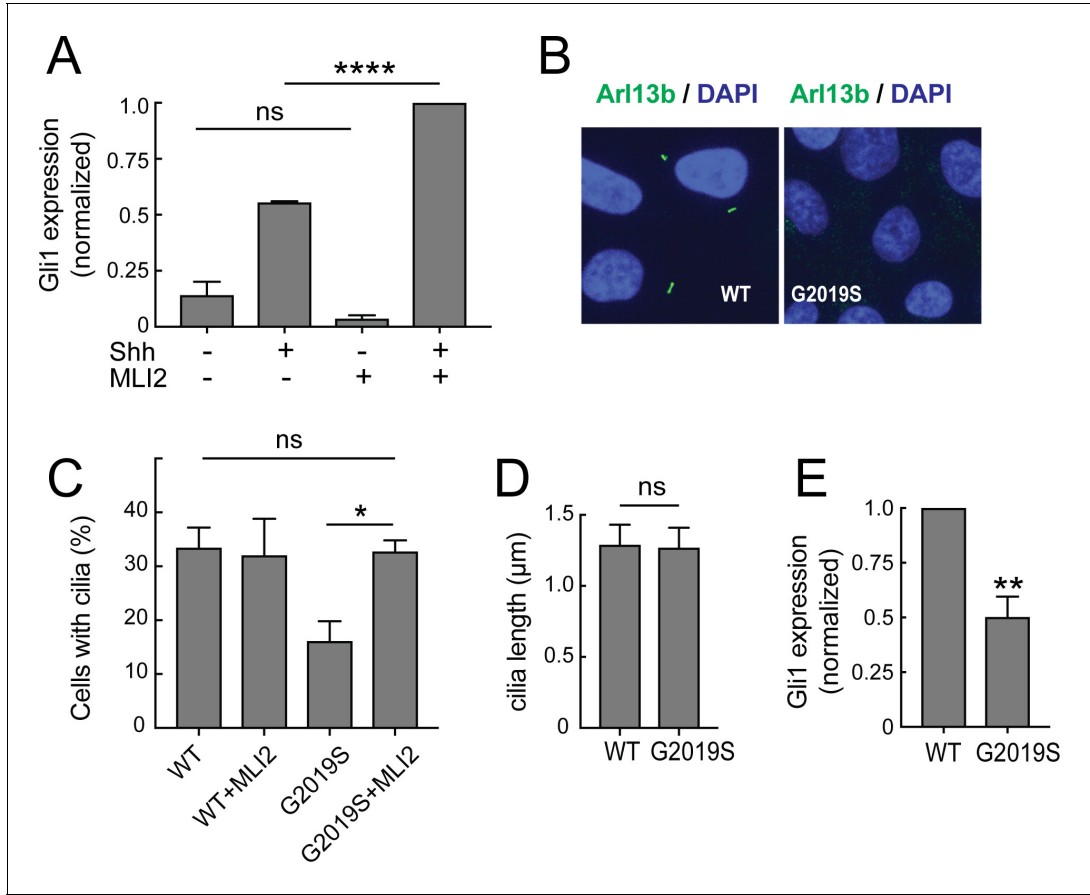

**Figure 7.** Ciliary signaling defects in MEFs and patient-derived iPS cells. (**A**) Confluent LRRK2 R1441G MEF cells were treated with 25 nM Sonic hedgehog (Shh) or PBS along with 200 nM MLi2 or DMSO for 24 hr under serum starvation conditions and then lysed for RNA isolation. Shown is the Gli1 mRNA level relative to GAPDH mRNA, normalized to control samples treated with both MLi2 and Shh. Error bars indicate standard error of the mean from three independent experiments. (**B–E**) Undifferentiated, patient derived LRRK2$^{G2019S/WT}$ iPS cells were cultured for 48 hr on matrigel coated coverslips. They were fixed and stained for Arl13b (**B**) or lysed for RNA isolation and qPCR analysis (**E**). (**B**) Arl13b (green) and DAPI (blue) staining in G2019S or wild type cells as indicated. (**C,D**) Quantitation of ciliation and cilia lengths for LRRK2$^{WT/WT}$ and LRRK2$^{G2019S/WT}$ iPS cells; MLi2 treatment was for 24 hr (400 nM). (**E**) Relative mRNA expression of human Gli1 in LRRK2$^{G2019S/WT}$ iPS cells normalized to LRRK2$^{WT/WT}$ corrected cells as indicated. Error bars indicate SEM from two independent experiments, each done in duplicate. P values were from Student's unpaired 2-tailed t test; ns = not significant, **,<0.01, ****,<0.001.
DOI: https://doi.org/10.7554/eLife.40202.013

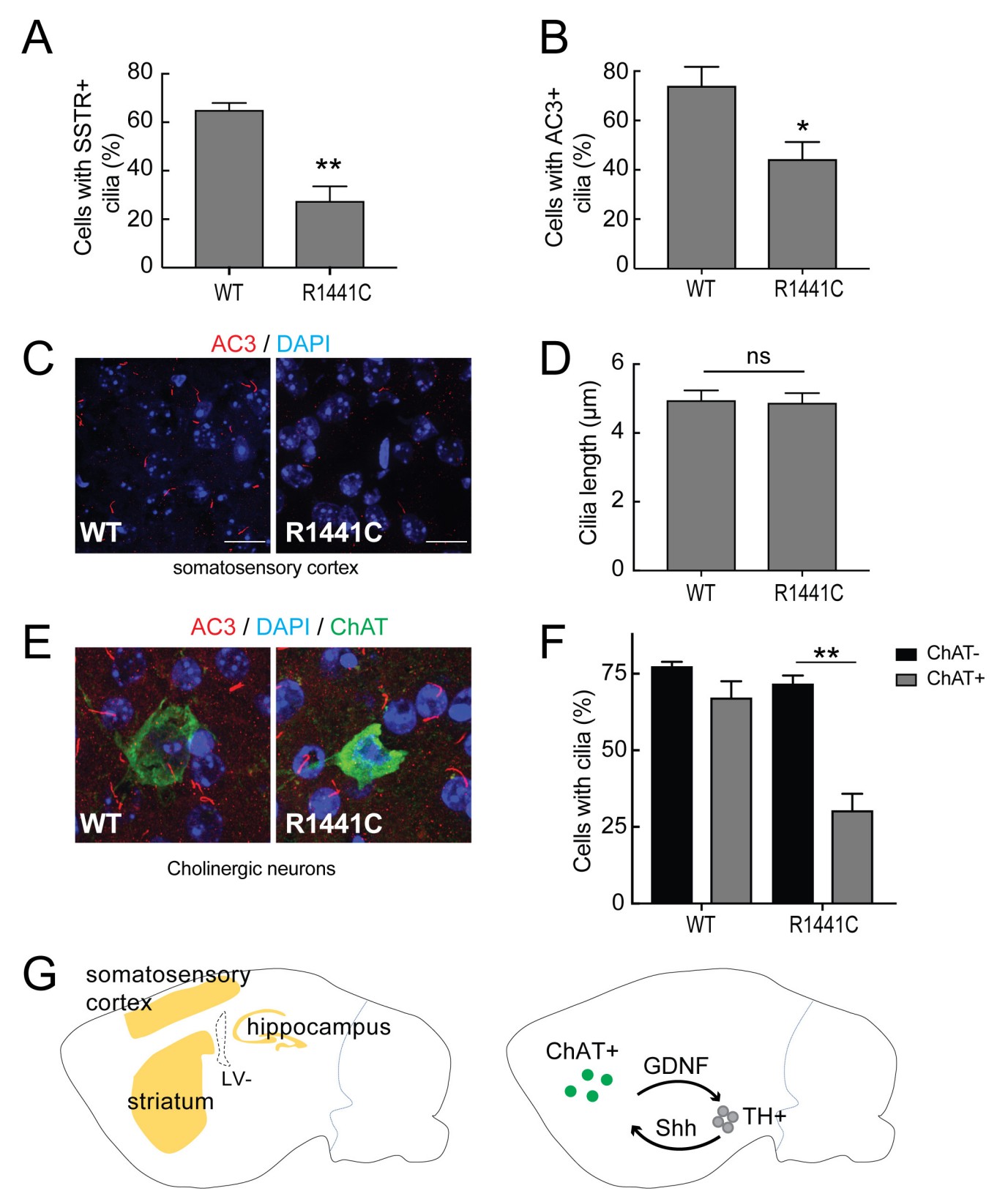

**Figure 8.** Primary cilia in mouse brain. Brains from wild type and 7 month old R1441C LRRK2 mice were stained with rabbit antibodies to neuron-specific primary cilia markers: somatostatin receptor 3 (SSTR3) or adenylate cyclase 3 (AC3). Primary cilia were quantified from somatosensory cortex,
*Figure 8 continued on next page*

*Figure 8 continued*

hippocampus and striatal regions. (**A**) Quantitation of neurons with SSTR3$^+$ primary cilia in the cortex. (**B**) Percent of total neurons with AC3$^+$ primary cilia in the cortex. (**C**) Representative image showing neuronal primary cilia stained for AC3 (red) and nuclei (blue) in the cortex. (**D**) Average length of primary cilia in the cortex (µm). (**E**) Cholinergic neurons in the striatum stained with goat anti-ChAT (choline acetyltransferase antibody, green), anti-AC3 (red) and nuclei (blue). (**F**) Quantitation of ChAT$^+$ and ChAT$^-$ neurons in the striatum with AC3$^+$ primary cilia;>50 cholinergic neurons were counted. (**G**) Cartoon depicting the regions in sagittal sections of mouse brains (yellow) where neurons were sampled for primary cilia (left image). Right panel depicts reciprocal communication between ChAT +neurons in the striatum (green) with tyrosine hydroxylase positive (TH+) dopaminergic neurons (grey) linked via Glia derived neurotrophic factor (GDNF) and Sonic hedgehog (Shh) signaling. Error bars represent standard error of the mean from n = 4 mice,>500 cells per mouse; p values were determined by Student's unpaired 2-tailed t-test.

DOI: https://doi.org/10.7554/eLife.40202.014

The following figure supplement is available for figure 8:

**Figure supplement 1.** Neuronal primary cilia in hippocampus and striatum.

DOI: https://doi.org/10.7554/eLife.40202.015

---

brains had primary cilia compared to wild type brains, detected using anti-SSTR3 and AC3 antibodies (*Figure 8A,B*). No significant difference was found between R1441C and wild type neurons in terms of overall cilia lengths in the cortex, hippocampus or striatum (*Figure 8D*). Cilia densities were slightly lower in R1441C hippocampus but the differences were not statistically significant (*Figure 8— figure supplement 1*).

Choline acetyl transferase (ChAT)-positive cholinergic neurons represent 1–4% of all neurons in the striatum (*Lim et al., 2014*). These cholinergic neurons send long-distance projections onto tyrosine hydroxylase positive, dopaminergic neurons and also receive signals from them (*Threlfell et al., 2010*; *Ding et al., 2006*). Dopaminergic neurons send Shh signals to nearby striatal, cholinergic neurons that in return, send neuroprotective signals back to the dopaminergic neurons (*Gonzalez-Reyes et al., 2012*). In the striatum, the overall ciliation of neurons was not significantly different in R1441C mice compared with wild type brains (*Figure 8F*). Importantly, when primary cilia were specifically monitored for ChAT-positive neurons, there was a striking reduction in the number of ciliated cholinergic neurons in R1441C striatum compared to wild type striatum (*Figure 8E,F*). R1441C mice had a highly significant cilia defect in cholinergic neurons that was not seen in wild type mice brains. These data suggest that loss of dopaminergic neurons in LRRK2-mediated Parkinson's disease may be a result of defective ciliogenesis in striatal cholinergic neurons, thus impairing a critical protective loop involving non-cell autonomous Shh signaling between cholinergic and dopaminergic neurons.

## Discussion

We have shown here that Rab8A and Rab10 have opposite effects on ciliogenesis in multiple cell types: Rab8A stimulates cilia formation, while Rab10 inhibits this process. Rab8A and Rab10 share numerous effector proteins, and their phosphorylation by pathogenic LRRK2 kinase interferes with the ability of most of these effectors to interact with the Rabs. However, a few effector proteins, including RILPL1, bind Rab8A and Rab10 with preference for their phosphorylated forms (*Steger et al., 2017*). This unusual class of effectors gains special focus in efforts to understand the downstream, physiological consequences of hyperactivated LRRK2 kinase that underlies familial Parkinson's disease.

We have shown here that pRab10 colocalizes with its RILPL1 effector on pericentriolar membranes adjacent to mother centrioles, and like Rab10, exogenous RILPL1 expression interferes with cilia formation. RILPL1 expression leads to more concentrated localization of pRab10 at centrosomes, consistent with the interaction of these proteins in cells. All of the above mentioned experiments involve exogenous expression of proteins that each may titrate out key players needed for cilia formation. Importantly, under conditions of *endogenous* protein expression, pathogenic LRRK2 kinase interferes with ciliation by generating pRab10 that binds endogenous RILPL1 and blocks cilia formation (*Figure 6D*). Moreover, LRRK2-mediated ciliation defects were not seen in cells lacking Rab10 or RILPL1, highlighting the importance of this pathway and these proteins in pathogenic LRRK2 signaling. How complexes of pRab10 block cilia formation together with RILPL1 protein will be important to elucidate in the future.

*Sato et al., 2014* showed previously that mice lacking Rab8A and Rab8B show normal primary cilia formation; these mice die, however due to microvillus atrophy caused by failure to correctly generate these intestinal, apical membranes. Additional knockdown of Rab10 in double-knockout MEF cells reduced the percentage of ciliated cells in that study. It is important to note Rab8B still present in our cells may explain the differences in ciliation phenotypes that we have reported here.

How does altering ciliogenesis relate to Parkinson's disease pathology? Parkinson's disease is associated with loss of dopaminergic neurons in the brain. Little is known about primary cilia in the adult brain; both neurons and astrocytes are ciliated, and they contain typical signaling molecules that include AC3, SSTR and the Smoothened protein that is important for Shh signaling. We found, for the first time, that cholinergic neurons in the striatum show cilia defects in R1441C LRRK2 mutant mice; defects were also seen in the somatosensory cortex, marked by either SSTR or AC3. Kottman and coworkers (*Gonzalez-Reyes et al., 2012*) have described a neuroprotective circuit in which Shh increases the resilience of dopaminergic neurons to toxins that mimic Parkinson's disease-associated, dopaminergic neuron loss. Cholinergic neurons of the striatum receive a Shh signal from dopaminergic neurons, which triggers secretion of dopaminotrophic, neuroprotective GDNF. Our data support a model in which pathogenic LRRK2 in cholinergic neurons impairs their ability to generate cilia, thereby blocking their receipt of the Shh signal from dopaminergic neurons. This pathway may be critical for the protection of dopaminergic neurons in aging brains, and represents a mechanism to explain the selective loss of dopamine neurons associated with the presentation of Parkinson's disease.

We cannot yet comment on the status of astrocyte cilia in LRRK2 mutant mice. Anti-AC3 antibodies rarely label astrocytes after postnatal day 10, and Arl13b has been used previously as an astrocyte cilia marker (*Kasahara et al., 2014*). In our hands, mouse anti-Arl13b antibodies labeled cell bodies and processes of GFAP-positive and negative glial cells but failed to label primary cilia specifically. A possible explanation is that astrocyte cilia are four times shorter than neuronal cilia in rodent brains and they are deeply recessed into a ciliary pocket, perhaps making them harder to label and/or detect (*Breunig et al., 2008*). Additional work will be needed to assess the consequences of pathogenic LRRK2 on astrocyte ciliiogenesis throughout the brain.

Finally, it is important to note that LRRK2-mediated familial Parkinson's disease is quite distinct from more classic diseases of ciliogenesis called ciliopathies (*Reiter and Leroux, 2017*). Patients carrying mutations in genes critical for cilia formation suffer diverse phenotypes including polycystic kidney disease, retinal degeneration, obesity, skeletal malformations and brain anomalies. Many of these symptoms are due to critical, developmental defects in Hedgehog and Wnt signaling processes that require ciliary pathways for proper signal transduction, even before birth. Because LRRK2 modulates Rab GTPase activity rather than permanently inactivating it, many cell types may have normal cilia formation or shorter cilia with decreased but still functional signaling capacity. It will be important to determine which cells and tissues display the greatest cilia defects in LRRK2 models of Parkinson's disease, and try to understand how those changes lead to specific loss of dopaminergic neurons in the brain.

In summary, pathogenic LRRK2 has a profound effect on cilia in cultured cells, and shown here, in human iPS cells and mutant mouse brains. Loss of ciliation correlates directly with loss of Shh signal reception in the striatum; other signaling events may also by altered. Olfactory receptor signals require cilia, and anosmia (loss of smell) is one of the earliest symptoms of Parkinson's disease. Future characterization of brain cilia in specific neuronal subtypes will provide critical information regarding pathways that may be altered in cells expressing pathogenic LRRK2; moreover, ciliation should be evaluated in cells from patients with idiopathic Parkinson's disease. Finally, Rab10 expression levels were reported as significantly increased in Alzheimer's disease brain (*Alzheimer's Disease Neuroimaging Initiative et al., 2017*), suggesting that Rab10's role as a ciliogenesis inhibitor may also be relevant to Alzheimer's disease pathology.

# Materials and methods

| Reagent type (species) or resource | Designation | Source or reference | Identifiers | Additional information |
|---|---|---|---|---|
| Gene (*Homo sapiens*) | RILPL1 | PMID: 29125462 | | |
| Gene (*Homo sapiens*) | LRRK2 | PMID: 2912546 PMID: 29125462 | NM_198578.3 | |
| Gene (*Homo sapiens*) | Rab8a | PMID: 29125462 | | |
| Gene (*Homo sapiens*) | Rab10Rab10 | PMID: 29125462 | | |
| Strain, strain background (*Mus musculus*) | B6.Cg-*Lrrk2tm1. 1Shn*/J mouse, | Jackson Laboratory | JAX stock #009346 | *Tong et al., 2009* |
| Strain, strain background (*Rattus norvegicus*) | SA Sprague Dawley rat | Charles River | Crl:SD/code 400 | *Foo et al., 2011* |
| Genetic reagent (shRNA) / (*Homo sapiens*) | human Rab10 | TRCN0000029191 | | |
| Genetic reagent (shRNA) / (*Homo sapiens*) | human Rab8a | TRCN0000300539 | | |
| Genetic reagent (shRNA) / (*Homo sapiens*) | human RILPL1 | TRCN0000159762 | | |
| Genetic reagent (shRNA) / (*Mus musculus*) | mouse RILPL1 | TRCN0000033243 | | |
| Genetic reagent (shRNA) / (*Mus musculus*) | mouse Rab8a | TRCN0000100422 | | |
| Cell line (*Homo sapiens*) | A549 Rab8 KO | PMID: 29125462 | | |
| Cell line (*Homo sapiens*) | A549 Rab10 KO | PMID: 29125462 | | |
| Cell line (*Homo sapiens*) | A549 | PMID: 29125462 | | |
| Cell line (*Homo sapiens*) | HEK293T | https://mrcppure agents.dundee.ac.uk/ | | |
| Cell line (*Mus musculus*) | MEF-R1441G | https://mrcppurea gents.dundee.ac.uk/ | | |
| Cell line (*Homo sapiens*) | hTert-RPE | ATCC | | |
| Cell line (*Homo sapiens*) | HeLa | ATCC | | |
| Cell line (*Homo sapiens*) | human iPSC | PMID: 24148854 | | *Sanders et al., 2014* |
| Recombinant DNA reagent | pMCB306-eGFP-Rab8 | PMID: 29125462 | | *Steger et al., 2017* *Steger et al., 2017* |
| Recombinant DNA reagent | pMCB306-eGFP-Rab10 | PMID: 9212815 | | *Purlyte et al., 2018* |
| Recombinant DNA reagent | pMCB306-eGFP-LRRK2 | PMID: 9212815 | | *Purlyte et al., 2018* |
| Recombinant DNA reagent | pENTR-2x-myc-LRRK2 | Addgene | 25361 | |
| Recombinant DNA reagent | pcDNA5D-FRT-TO-RILPL1 | PMID: 29125462 | | *Steger et al., 2017* *Steger et al., 2017* |

*Continued on next page*

*Continued*

| Reagent type (species) or resource | Designation | Source or reference | Identifiers | Additional information |
|---|---|---|---|---|
| Transfected construct (gRNA) | RILPL1 | https://mrcppureagents.dundee.ac.uk/ | DU57861 | |
| Transfected construct (gRNA) | RILPL1 | https://mrcppureagents.dundee.ac.uk/ | DU57867 | |
| Antibody | rabbit monoclonal anti-RILPL1 | Sigma | HPA-014314 | (1:500) |
| Antibody | rabbit monoclonal anti-Rab10 | Cell Signaling | 8127 | (1:1000) |
| Antibody | rabbit monoclonal anti-LRRK2 | Abcam | AB133518 | (1:1000) |
| Antibody | rabbit monoclonal anti-phospho T73 Rab10 | Abcam | AB230261 | (1:2000) |
| Antibody | rabbit monoclonal anti-Rab8 | Cell Signaling | 6975 | (1:200) |
| Antibody | mouse monoclonal anti-myc | 9E10 clone hybridoma | | (1:1) |
| Antibody | chicken polyclonal anti-GFAP | Novus Biologicals | NBP1-05198 | (1:5000) |
| Antibody | goat polyclonal anti-choline acetyl transferase | Millipore | AB144P | (1:200) |
| Antibody | mouse monoclonal anti-Arl13b | Neuromab | 75–287 | (1:2000) |
| Antibody | mouse monoclonal anti-Cep164 | Santa Cruz | SC-515403 | (1:200) |
| Antibody | rabbit polyclonal anti-adenylate cyclase 3 | Santa Cruz | SC-588 | (1:100) |
| Antibody | mouse monoclonal anti-NeuN | Biolegend | 834501 | (1:2000) |
| Antibody | mouse monoclonal anti-GFP | Neuromab | N86-38 | (1:2000) |
| Chemical compound, drug | Lipofectamine 3000 | Life Technologies | | |
| Chemical compound, drug | MLi2 | PMID: 29125462 | | |
| Software, algorithm | FIJI | | | |

## Reagents

MLi2 LRRK2 inhibitor (*Scott et al., 2017*) was synthesized as described in (*Miller et al., 2014*). All DNA constructs and antibodies generated for the experiments are described in (*Purlyte et al., 2018*) and can be found at https://mrcppureagents.dundee.ac.uk.

## General cloning and plasmids

DNA constructs were amplified in *E. coli* DH5α and purified using mini prep columns (Econospin). DNA sequence verification of all plasmids was performed by Sequetech (http://www.sequetech.com). The following constructs were used: eGFP-Rab8A and eGFP-Rab10 were cloned into lentivirus vector pSLQ1371 between BstB1 and AvrII sites (*Purlyte et al., 2018*). Rab8A T72E/A and Rab10 T73E/A were generated by site directed mutagenesis using Pfu polymerase (Agilent Technologies). eGFP-LRRK2-G2019S was cloned into modified pSLQ1371 with eGFP at the N-terminus. Myc-LRRK2 was obtained from Addgene (#25361) and the R1441G mutation was introduced as described (*Purlyte et al., 2018*). RILPL1 was cloned into pcDNA5d/FRT/TO using Gibson assembly and RILPL1-C and RILPL1-N were made by site directed mutagenesis. Lentiviral shRNA pLKO.1 plasmids were obtained as bacterial stocks from Sigma: mouse Rab8A (TRCN0000100422), human Rab8A (RCN0000300539), human Rab10 (TRCN0000029191), mouse Rab10 (TRCN0000335543), mouse RILPL1 (TRCN0000033243) and human RILPL1 (UTR TRCN0000159762). For human Rab10, pSIH vector was also used with a GFP reporter. The shRNA sequence for hRab10 5'- GATCCGAAGA TCAAGCTACAGATCTTCCTGTCAGAATCTGTAGCTTGATCTTCTTTTTG was cloned using *EcoRI* and *BamHI* restriction sites.

## Cell culture, transfections and ciliogenesis

HEK293T, HeLa, A549 and hTERT-RPE cells were obtained from ATCC and were cultured in Dulbecco's modified Eagle's medium containing 10% fetal bovine serum, 2 mM Glutamine, penicillin (100 U/ml)/streptomycin (100 µg/ml) and non-essential amino acids. HEK293T cells were transfected with Polyethylenimine HCl MAX 4000 (PEI) (Polysciences, Inc.) as described (Reed et al., 2006). RPE and HeLa cells were transfected with Fugene6 (Promega) while rat astrocytes and A549 cells were transfected using lipofectamine 3000 according to the manufacturer. Cells were checked routinely for Mycoplasma using either MycoALert Mycoplasma Detection Kit (Lonza LT07-318) or PCR.

## Human iPS and rat astrocyte cell culture

Human patient derived G2019S = NM_198578.3 (*LRRK2*): c.6055G > A (p.Gly2019Ser) iPS cultures were maintained as described (*Sanders et al., 2014*). Cells were plated on 4% matrigel coated, 24 well plates. Colonies of iPS cells were dissociated with Accutase, and resuspended in mTesR medium with 10 µM Y-27632. Cultures of primary rat astrocytes were obtained by antibody panning from rat pups as described (*Foo et al., 2011*). In brief, six to ten postnatal Sprague-Dawley rat cortices were mechanically and enzymatically dissociated to produce single cells. They were passed over successive, antibody-coated negative panning plates to rid the suspension of microglia, endothelial cells, and oligodendrocyte precursor cells before selecting for astrocytes with an anti-ITGB5-coated plate. Astrocytes that attached to the anti-ITGB5-coated plate were trypsinized and plated onto poly-D lysine coated ACLAR coverslips. Cells were incubated at 37°C with 10% $CO_2$ and 90% $O_2$ for one to three weeks.

## Lentivirus production

Lentivirus-based shRNA was used for gene knockdown of human Rab8A, mouse/human Rab10, mouse/human RILPL1 using pLKO or pSIH-GFP vector. The lentiviral vector was co-transfected with packaging vectors psPAX2, pMD2 VSV-G in HEK293T cells using PEI. pLKO-puro-scramble or pSIH-GFP-scramble was used as control. After 48 hr, culture supernatants were collected and virus concentrated 10x overnight with Lenti-X concentrator (Clontech) according to the manufacturer. Virus was transduced onto target cells with polybrene (2 µg/ml). To make stable cells, infected cells were selected with puromycin (0.2–1 µg/ml) 48 hr after infection. Expression of the target protein was verified by GFP-fluorescence, qPCR and/or immunoblotting.

## Lysis and immunoblotting

Cells were lysed 24 hr after transfection in ice-cold lysis buffer containing 50 mM Tris/HCl, pH 7.5, 1% (v/v) Triton X-100, 1 mM EGTA, 1 mM sodium orthovanadate, 50 mM NaF, 10 mM 2-glycero-phosphate, 5 mM sodium pyrophosphate, 0.1 µg/ml mycrocystin-LR (Enzo Life Sciences), and EDTA-free protease inhibitor cocktail (Sigma). Lysates were centrifuged at 13800 x $g$ for 15 min at 4°C and supernatants were quantified by Bradford assay (Thermo Scientific) and 60 µg of proteins used for immunoblotting. Antibodies were diluted in blocking buffer containing 5% milk in 0.05% Tween in Tris buffered saline (TBS) and incubated overnight on the blots.

## Inducing primary cilia

Cells were grown to confluency in DMEM with serum. Mouse embryonic fibroblasts (MEFs) (WT and R1441G) as well as hTERT-RPE cells were ciliated by overnight serum starvation in DMEM medium alone. For A549 cells, cells were plated at 80% confluency and 24 hr later, subjected to 2% serum for 48 hr. Neither rat astrocytes nor iPS cells needed cilia induction. Rat astrocytes made primary cilia after one week in culture while undifferentiated iPS cells made cilia by 72 hr in culture on matrigel. To identify primary cilia, cells were stained for cilia markers Arl13b or acetylated tubulin.

## Membrane fractionation

Cells were chilled on ice, washed with ice cold PBS, and swelled in 10 mM HEPES pH 7.4. After 15 min, 5X buffer was added to achieve a final concentration of resuspension buffer (50 mM HEPES pH 7.4, 150 mM NaCl, 2 mM MgCl$_2$, 2 mM DTT, 20 µM GDP, 1X protease inhibitor cocktail (Sigma)), and the suspension was passed 20 times through a 25 gauge syringe. Nuclei were pelleted by centrifugation at 1,000 X $g$ for 5 min at 4°C. The postnuclear supernatant was spun 100,000 X $g$ for 20 min in a table top ultracentrifuge in TLA100.2 rotor; the resulting supernatant was the cytosol fraction. Membrane pellets were solubilized in 1% Triton X100 containing 1X resuspension buffer. Protein was estimated by Bradford assay (BioRad, Richmond, CA). Samples containing 25 µg of membrane protein, or the equivalent volume of cytosolic protein, were heated at 95°C for 2 min after addition of 5X SDS–PAGE sample buffer. Samples were loaded in duplicate onto 12% SDS-PAGE gels for imunoblotting; filters were incubated with chicken anti-GFP (Aves, 1:2000) overnight then with donkey anti-chicken 680 (Licor, 1:10,000) for 1 hr, imaged using an Odyssey Infrared scanner (Licor), and quantified using ImageJ software.

## Hedgehog signaling

MEFs were cultured in 24 well plates, and upon confluency, serum starved in DMEM and treated with 200 nM MLi2 or DMSO. At the same time, cells were treated with either 25 nM Sonic hedgehog (Shh) or PBS. After 16 hr ± MLi2,±Shh treatment, cells were lysed for RNA extraction and qPCR of Gli1 mRNA. Human patient derived iPS cells already have high Gli1 levels and did not need Shh treatment; iPS cells were lysed for qPCR analysis of Gli1 after 72 hr.

## Quantitative PCR

To extract total RNA, cells are lysed in Trizol (Sigma) for 5 min in an RNA-free zone. RNA extraction was performed using chloroform/isopropyl alcohol according to the manufacturer. RNA was stored at −80°C. cDNA was synthesized using 500 ng to 1 µg RNA using high capacity cDNA synthesis kit with multiscribe reverse transcriptase (Applied Biosystems). The cDNA was diluted 50 fold and 1 µl cDNA was used as template in the PowerUp SYBR green 2X master mix and analyzed in a qPCR machine (7900HT Applied Biosystems). Gene expression was compared using 2(delta-delta Ct) method with GAPDH or Actin as internal controls. The primers used for the qPCR were as follows – RAB10 HGNC:9759 -fw-GTGGGGAAGACCTGCGTCCTTT, rv- GAGGTTGTGATGGTGTGAAATCGC; RAB8A HGNC:7007 -fw–CTGGCGAGAGTGAAAAATGC, rv–AAAAGCTGGCCCTCGACTAT; RILPL1 HGNC:26814-fw–ATGGAAGAGGAAAACCGAATACC, rv–AGGCGCTTCTTATCTCGGGA; ACT HGNC:929-fw–GGCATCCTCACCCTGAAGTA,rv–AGAGGCGTACAGGGATAGCA; HPRT1 HGNC:5157-fw–GGTCAGGCAGTATAATCCAAAG, rv–GGACTCCAGATGTTTCCAAAC; GLI1 HGNC:4317 -fw–GAAGTCATACTCACGCCTCGAA, rv–CAGCCAGGGAGCTTACATACAT; Mus musculus GAPDH-fw–AGGTCGGTGTGAACGGATTTG, rv–TGTAGACCATGTAGTTGAGGTCA; Mus musculus RAB10-fw–GGCAAGACCTGCGTCCTTTT, rv–GTGATGGTGTGAAATCGCTCC; mRILPL1-fw–

GGTTGCGCGTAGAGAGGATG, rv–CTCACGCTCTGACATGCCTTC; Mus musculus GLI1-fw-CCAAGCCAACTTTATGTCAGGG, rv- AGCCCGCTTCTTTGTTAATTTGA.

## Generation of mutant mice

The LRRK2$^{R1441C/R1441C}$ mice were obtained from The Jackson Laboratory (B6.Cg-*Lrrk2*$^{tm1.1Shn}$/J mouse, JAX stock #009346; *Tong et al., 2009*) and kept in specific pathogen-free conditions at the University of Dundee (UK). All animal studies were ethically reviewed and carried out in accordance with Animals (Scientific Procedures) Act 1986 and regulations set by the University of Dundee and the U.K. Home Office. Animal studies and breeding were approved by the University of Dundee ethical committee and performed under a U.K. Home Office project license. Mice were multiply housed at an ambient temperature (20–24°C) and humidity (45–55%) and maintained on a 12 hr light/12 hr dark cycle, with free access to food (SDS RM No. three autoclavable) and water. Genotyping of mice was performed by PCR using genomic DNA isolated from ear biopsies. Primer 1 (5' -CTGCAGGC TACTAGATGGTCAAGGT −3') and Primer 2 (5' –CTAGATAGGACCGAGTGTCGCAGAG- 3') were used to detect the wild-type and knock-in alleles. Homozygous LRRK2-R1441C and littermate wild-type controls (7 months of age) were used for experiments shown in *Figure 8*. The genotype of these mice was confirmed by PCR on the day of experiment.

## Generation of knock out A549 cells

A549 Rab8A knockout (*Steger et al., 2017*) and Rab10 knock-out (*Ito et al., 2016*) cell lines have been described (*Steger et al., 2017*). The A549 RILPL1 knockout cell line was generated using sense and anti-sense paired guides DU57861 and DU57867 (available at https://mrcppureagents.dundee.ac.uk), targeting exon 3 of RILPL1; sense guide: GGTGATGAAGAAGCTGAAGG (DU57861); anti-sense guide: GCTCCCGCTCTGACATGCCT (DU57867). Cells at about 80% confluency were co-transfected in a six-well plate with the pair of constructs using Lipofectamine LTX reagent (9 µl Lipofectamine LTX and 2.5 mg DNA per well). After 24 hr transfection, the medium was replaced with medium Supplemented with puromycin (2 mg/ml). After 24 hr selection, the medium was replaced with medium without puromycin and cells were left to recover for 48 hr before performing single-cell sorting using an Influx cell sorter (Becton Dickinson). Single cells were placed in individual wells of a 96-well plate containing DMEM Supplemented with 10% FBS, 2 mM L-glutamine, 100 units/ml penicillin,100 mg/ml streptomycin and 100 mg/ml normocin (InvivoGen). At about 80% confluency individual clones were transferred into six-well plates and screened for RILPL1 knock-out by immunoblotting.

## Light microscopy

Cells were plated on collagen coated coverslips transfected with indicated plasmids. Cells were fixed with 3% paraformaldehyde for 20 min, permeabilized for 3 min in 0.1% Triton X 100 (or 0.1% saponin for anti-phospho Rab10 antibody staining) and blocked with 1% BSA in PBS. Antibodies were diluted as follows: mouse anti-Arl13b (1:1000, Neuromab), mouse anti-GFP (Neuromab, 1:1000); mouse anti-Myc (9E10 Hybridoma culture supernatant – 1:2), rabbit anti-RILPL1 (Sigma, 1:500), rabbit anti-GCC185 (Cheung et al. 2015, 1:1000), rabbit anti phospho Rab10 (1:1000, Abcam), rabbit anti adenylate cyclase III (Santa Cruz 1:100), mouse anti-NeuN (Proteintech, 1:1000), goat anti choline acetyltransferase (Millipore 1:100), rabbit anti Rab8A and Rab10 (1:1000, Cell Signaling), rabbit anti RILPL1 antibody (*Steger et al., 2017*), mouse anti tubulin (1:2000, Sigma), mouse anti acetylated tubulin (1:2000, Sigma), mouse anti Centrin-3 (gift from Tim Stearns, Stanford), chicken anti GFAP (1:5000, Novus Biologicals), mouse anti Transferrin receptor (1:500, BD Bioscience). Highly cross absorbed H + L secondary antibodies (Life Technologies) conjugated to Alexa 488, Alexa 568 or Alexa 647 were used at 1:2000 or 1:4000. All antibody dilutions for tissue staining included 1% DMSO to help antibody penetration. All images were obtained using a spinning disk confocal microscope (Yokogawa) with an electron multiplying charge coupled device (EMCCD) camera (Andor, UK) and a 100 × 1.4 NA oil immersion objective. Images were analyzed using Fiji (https://fiji.sc/) and whenever necessary, were presented as maximum intensity projections. Nuclei were stained using 0.1 µg/ml DAPI (Sigma).

## In vitro prenylation

For in vitro prenylation, cytosol obtained from membrane fractionation was incubated with 6.7 µM biotin-geranyl pyrophosphate (Jena Bioscience) for 4 hr, rotating at room temperature. GFP-Rab8A or GFP-Rab10 were then immunoprecipitated using GFP binding protein immobilized on Sepharose for 1 hr at 4°C. Resin was washed 3 times with 1X membrane fractionation resuspension buffer and eluted with 2X SDS-PAGE sample buffer. The elutions were run on 12% SDS-PAGE gels for immunoblotting, detecting prenylation Streptavidin 800 nm (Licor, 1:1000) then chicken anti GFP (Aves 1:2000). Membranes were incubated with donkey anti chicken 680 (Licor 1:10,000) for 1 hr, imaged using an Odyssey Infrared scanner (Licor0 and quantified using ImageJ software. For Rab8A, dishes were treated with 10 µM lovastatin 16 hr before lysis—because Rab8A is monoprenylated, this increased the pool of non-prenylated Rab8A in the cytosol, enabling us to assay for prenylation capacity in vitro.

## Mouse brain processing and primary cilia analyses

Homozygous LRRK2-R1441C and littermate wild-type controls (7 months of age) were terminally anesthetized and perfused by injection of PBS into the left cardiac ventricle followed by injection of 4% paraformaldehyde in PBS before collecting the brains for analysis. Mice brains were immersed in 30% sucrose for 2 days until they were completely submerged. Brains were then embedded in plastic blocks with OCT (BioTek, USA), frozen on dry ice and stored at −80°C. While embedding, the anterior and posterior sides of the brains in the block were noted. For cryosectioning, OCT blocks were kept at −20°C for 10 min. The brains were oriented to cut sagittal sections on the cryotome (Leica CM3050S, Germany) at 16 µm thickness and adhered onto SuperFrost plus tissue slides (Thermo Fisher, USA). This orientation helped in visualizing cortex, hippocampus and striatal regions (*Figure 8G*).

Slices closest to the midline were chosen for immunostaining. For primary cilia staining in brain sections, frozen slides were thawed at RT for 10 min then gently washed with PBS for 5 min. Sections were permeabilized with 0.1% Triton X-100 in PBS at room temperature for 15 min. Sections were blocked with 1% bovine serum albumin and 2% bovine calf serum in PBS for 2 hr at RT. Sections were incubated overnight with primary rabbit polyclonal antibodies and incubated with secondary antibodies (Thermo Scientific, 1:1000) at room temperature for 2 hr. Stained tissues were overlayed with Mowiol and a glass coverslip. Three brain areas were chosen – Somatosensory area of the cerebral cortex, pyramidal layer of the hippocampus and striatal area of the caudoputamen, adjacent to the lateral ventricle. Four to five regions within each brain area were imaged using 0.75 µm Z-sampling for each mouse brain. All image visualizations and analyses were performed using Fiji (https://fiji.sc). To analyze primary cilia, maximum intensity projections of Z-stacks were median filtered, thresholded and the resulting mask was subjected to the skeletonize function. Number of Primary cilia structures in these masks were then counted using Analyze > Analyze Particles. To measure primary cilia lengths, the output image from the previous step was subsequently analyzed using Analyze > Skeleton > Analyze Skeleton > Longest shortest path function. The number of nuclei in the cerebral cortex and striatal regions of the brain were determined by analyzing maximum intensity projections of the DAPI channel. The image was median filtered, subjected to 'Fill Holes', 'Watershed' followed by Analyze > Analyze Particles. The number of nuclei in the pyramidal layer of the hippocampus was determined manually by differential pseudo-coloring of the Z-stack to overcome counting errors caused by tight packing of the nuclei and intense nucleoli staining of neurons in this region. For cilia analysis of iPS cells, undifferentiated cells were plated on matrigel coated coverslips and cultured for 72 hr; the cells were subconfluent, two dimensional monolayers rather than organoids. Upon confluency (5 days), they lose their ciliation and were not analyzed further.

### Quantitation of centriolar distance

A549 cells transfected with RILPL1-GFP and stained for Centrin-3 or gamma tubulin were imaged on a confocal microscope with Z-step size of 0.5 µm. Distances were quantified from cells where only two dots were visible; cells in mitosis were excluded based on nuclear DAPI staining. Only a single plane from a Z-stack was used to quantify the distance between centrioles to eliminate error introduced by Z-sampling. The distance was measured using FIJI (https://fiji.sc) by drawing a straight line between the centers of the centriole dots. RILPL1 expression was quantified in FIJI by first

subtracting background from the sum-intensity projection images, outlining the cells, followed by measuring the integrated density of the signal from a single cell.

## Statistics

Graphs were made using Graphpad Prism six software. Error bars indicate SEM. Unless specified, a Student's T-test was used to test significance. Two tailed p-values<0.05 were considered statistically significant.

## Acknowledgements

This work was supported by grants from the Michael J Fox Foundation for Parkinson's research (to SRP and DRA), the Medical Research Council (grant MC_UU_12016/2 to DRA); and the U.S. National Institutes of Health DK37332 (to SRP). We are especially grateful to Gregor Bieri from the lab of Aaron Gitler (Stanford) for their culturing of and guidance for use of patient derived iPS cells.

## Additional information

### Competing interests

Suzanne R Pfeffer: Reviewing Editor, *eLife*. The other authors declare that no competing interests exist.

### Funding

| Funder | Grant reference number | Author |
|---|---|---|
| Michael J. Foundation for Parkinson's Research | LEAPS and RDI | Herschel S Dhekne<br>Izumi Yanatori<br>Rachel C Gomez<br>Francesca Tonelli<br>Federico Diez<br>Birgitt Schüle<br>Martin Steger<br>Dario R Alessi<br>Suzanne R Pfeffer |
| Medical Research Council | MC_UU_12016/2 | Dario R Alessi |
| National Institutes of Health | DK37332 | Suzanne R Pfeffer |

The funders had no role in study design, data collection and interpretation, or the decision to submit the work for publication.

### Author contributions

Herschel S Dhekne, Conceptualization, Data curation, Formal analysis, Validation, Investigation, Writing—original draft, Project administration, Writing—review and editing, Designed and conceived research, Analyzed data and wrote the paper, Performed the experiments; Izumi Yanatori, Conceptualization, Data curation, Formal analysis, Investigation, Writing—review and editing, Designed and conceived research, Analyzed data and wrote the paper, Performed the experiments; Rachel C Gomez, Investigation, Writing—review and editing, Designed and conceived research, Analyzed data and wrote the paper, Performed the experiments; Francesca Tonelli, Conceptualization, Investigation, Writing—review and editing, Designed and conceived research, Analyzed data and wrote the paper, Performed the experiments; Federico Diez, Investigation, Performed the experiments; Birgitt Schüle, Resources, Writing—review and editing, Generated key cell lines; Martin Steger, Conceptualization, Writing—review and editing, Designed and conceived research, Analyzed data and wrote the paper; Dario R Alessi, Conceptualization, Formal analysis, Supervision, Funding acquisition, Project administration, Writing—review and editing, Designed and conceived research, Analyzed data and wrote the paper; Suzanne R Pfeffer, Conceptualization, Data curation, Formal analysis, Supervision, Funding acquisition, Investigation, Visualization, Writing—original draft, Project

administration, Writing—review and editing, Designed and conceived research, Analyzed data and wrote the paper

## Author ORCIDs
Herschel S Dhekne (iD) https://orcid.org/0000-0002-2240-1230
Izumi Yanatori (iD) http://orcid.org/0000-0002-1945-5473
Rachel C Gomez (iD) http://orcid.org/0000-0002-6712-322X
Martin Steger (iD) https://orcid.org/0000-0003-1637-8190
Dario R Alessi (iD) https://orcid.org/0000-0002-2140-9185
Suzanne R Pfeffer (iD) https://orcid.org/0000-0002-6462-984X

## Ethics

Animal experimentation: All animal studies were ethically reviewed and carried out in accordance with Animals (Scientific Procedures) Act 1986 and regulations set by the University of Dundee and the U.K. Home Office. Animal studies and breeding were approved by the University of Dundee ethical committee and performed under a U.K. Home Office project license.

## Decision letter and Author response

Decision letter https://doi.org/10.7554/eLife.40202.018
Author response https://doi.org/10.7554/eLife.40202.019

## Additional files

### Supplementary files
• Transparent reporting form
DOI: https://doi.org/10.7554/eLife.40202.017

### Data availability
All data generated or analysed during this study are included in the manuscript and supporting files.

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
