## [Decision Letter]

[**Editorial note:** This article has been through an editorial process in which the authors decide how to respond to the issues raised during peer review. The Reviewing Editor's assessment is that all the issues have been addressed.]

Thank you for submitting your article "A pathway for Parkinson's Disease LRRK2 kinase to block primary cilia and Sonic hedgehog signaling in the brain" for consideration by *eLife*. Your article has been reviewed by three peer reviewers, including Christopher G Burd as the Reviewing Editor and Reviewer #1, and the evaluation has been overseen by Vivek Malhotra as the Senior Editor. The following individual involved in review of your submission has agreed to reveal their identity: Peter Novick (Reviewer #3). Reviewer #2 remains anonymous.

The Reviewing Editor has highlighted the concerns that require revision and/or responses, and we have included the separate reviews below for your consideration. If you have any questions, please do not hesitate to contact us.

Summary:

The manuscript presents an investigation into the roles of LRRK2, Rab8A, Rab10, and RILPL1 in ciliogenesis and report that Parkinson's Disease-associated alterations in LRRK2 activity lead to compromised cilium assembly and Hedgehog signaling. Overall, the reviewers considered the conclusion that LRRK2 regulates the activities of Rab8 and Rab10 in ciliogenesis to be well supported by the data. Although the study provides little mechanistic insight into the specific functions of Rab8 and Rab10 in ciliogenesis, it does provide an interesting and potentially important insight into the etiology of Parkinson's disease.

Major concerns:

1) It is not firmly established that Rab8 and Rab10 play important roles in ciliogenesis in cell types other than the A549 cell line (see reviewer 2, major comment 1), which is not commonly used for investigating ciliogenesis. The impact of the work could be better evaluated by repeating a limited number of key experiments in RPE cells or wildtype MEFs (e.g. to show that Rab10/RILPL1 can inhibit ciliogenesis in these cells). This seems to have been addressed with experiments done using RPE cells (subsection “Rab8 activates while Rab10 inhibits cilia formation in A549 cells”, first paragraph), but the data is not shown. At a minimum, this data should be shown.

2) Neither the Rab8 or Rab10 T-to-E/A mutant proteins are functional and it seems likely that the mutant Rabs are non-functional independent of their presumed effects on phosphorylation, and this point is not adequately explored or clearly conveyed. It is shown that Rab10 T73A is cytosolic, explaining why it is non-functional. Why are the other rab8 and Rab10 mutants non-functional? Are they isoprenylated? If the mutants do not selectively affect phosphorylation (e.g. they alter isoprenylation), their use seems tangential to the main point of the paper.

3) The effect of the Rab10 LRRK2 phosphorylation site mutations (T73A, E) on its association with RILP1 should be determined, similar to the analysis of Rab8 mutants and RILP1 (Figure 3—figure supplement 1). This information is important for interpreting the phenotypes of the mutant proteins.

4) Please note that each of the reviewers were confused by certain aspects of the presentation (see the reviews). I encourage you to consider revising the presentation to address these sources of confusion.

Separate reviews (please respond to each point):

*Reviewer #1:*

This study addresses the role of LRRK, a protein kinase implicated in Parkinson's disease, in ciliogenesis. Mutations in LRRK that cause PD activate its protein kinase activity, but how this leads to PD is unknown. The study follows an earlier publication that reported that Rab proteins are the principal substrates for LRRK, including Rab8 and Rab10, the focus of this study. Rab8 has previously been implicated in ciliogenesis, and the authors present evidence that phosphorylation of Rab8 by LRRK inhibits its function in ciliogenesis. In contrast, the authors find that depletion of Rab10, another Rab GTPase implicated in ciliogenesis, potentiates ciliogenesis and that phosphorylation of Rab10 by LRRK potentiates its inhibitory effect on ciliogenesis via enhanced interaction with RILPL1, a Rab10 effector of unknown function. It is shown that depletion of RILPL1 potentiates ciliogenesis, similar to depletion of Rab10. Finally, it is shown that PD mutations in LRRK cause a decrease in the proportion of ciliated cells and deficient SSH signaling in striatal cholinergic neurons of the mouse brain. The specific functions of Rab8 or Rab10 in ciliogenesis or PD are largely unknown. This is a technically sound, well controlled study, that clearly supports roles for Rab8 and Rab10 in ciliogenesis, though if these GTPases play direct, specific roles in ciliogenesis remains an open question. In my opinion, the major value of this study lies in the identification of functional consequences for LRRK-mediated phosphorylation of Rab8 and Rab10 in ciliogenesis, a finding that puts a spotlight on ciliogenesis for future studies of PD etiology.

Phenotypic analyses of Rab8a/b knockout mice by Sato et al., 2014, showed essentially no effect on cilia number and little evidence of ciliopathies that would be expected by the loss of a key factor in ciliogenesis. Moreover, they reported that knockout of Rab10 resulted in a loss of cilia, the opposite reported here. How do the authors reconcile these published observations with their results? If I understand correctly, the authors propose that Rab8 "activates" ciliogenesis, rather than provides an essential role, but it is unclear to me what "activate" means.

It is shown that loss of RILPL1 potentiates ciliogenesis, similar to loss of Rab10. It is also shown that Rab10 does not require RILPL1 for its effects on ciliogenesis ("This suggests that Rab8A and Rab10 do not require RILPL1 for their effects on ciliogenesis…"). So why are Rab10 and RILPL1 proposed ("LRRK2 kinase acts via a pRab10-RILPL1 pathway to inhibit ciliogenesis") to function on the same ciliogenesis pathway? It would be helpful if the authors clarified this conclusion.

Additional data files and statistical comments:

Rigour and extent of the statistical information are satisfactory.

*Reviewer #2:*

Here, Dhekne et al. investigate the roles of LRRK2, Rab8A, Rab10, and RILPL1 in ciliogenesis and report that Parkinson's Disease-associated alterations in LRRK2 activity lead to compromised cilium assembly and Hedgehog signaling, which may contribute to Parkinson's pathogenesis. This study builds on two recent reports in *eLife* from these groups (working also with Matthias Mann's group) that identified Rab8A and Rab10 as LRRK2 substrates, characterized phosphorylation-dependent binding of RILPL1 to these Rab's, and found reduced ciliogenesis in fibroblasts with elevated LRRK2 activity. In this manuscript, the authors investigate how Rab8A, Rab10, and RILPL1 control ciliogenesis downstream of LRRK2, finding that Rab8A promotes ciliogenesis (as previously reported), while RILPL1 promotes Rab10-dependent inhibition of cilium formation. Additionally, the authors show that increased LRRK2 activity leads to decreased reduced ciliogenesis (in MEFs, human iPS cells and mouse brain) and diminished cilium-dependent Hedgehog signaling (in MEFs and human iPS cells).

The negative regulation of ciliogenesis by Rab10/RILPL1 is an interesting finding that, if generalizable to other cell types/tissues, represents a novel insight into ciliogenesis that motivates future analysis of the underlying mechanism. Additionally, the findings that LRRK2 hyper-activity inhibits ciliogenesis and blunts Hedgehog signaling provide further evidence that ciliary defects may contribute to Parkinson's Disease. Lastly, the observation that Rab10 knockdown suppresses ciliogenesis defects seen in MEFs with elevated LRRK2 activity provides an important connection between the two aforementioned findings. As detailed below, the study could be strengthened by further examining Rab10's role in ciliogenesis in cell lines that are more commonly used to study cilia. Additionally, some data are presented in a confusing manner and do not support some of the conclusions reached. These unclear sections significantly obscure the key results and should be revised before publication in *eLife*.

1) Many of the experiments are done in A549 cells, but A549 cells ciliate at a relatively low rate (~25%) and are not commonly used to study ciliogenesis. Thus, it is possible that the inhibitory effects on ciliogenesis of Rab10/RILPL1 may be a distinct feature of this cell type rather than indicative of a more general role for these proteins in regulating cilium formation (especially as Sato et al., 2014, reported Rab10 to be instead a positive regulator of ciliogenesis that acts redundantly with Rab8A and Rab8B). This concern is partially mitigated by the data from LRRK2-R1441G MEFs shown in Figure 6, but it would nonetheless be helpful to repeat a limited number of key experiments in RPE cells or wildtype MEFs (e.g. to show that Rab10/RILPL1 can inhibit ciliogenesis in these cells). In the first paragraph of the subsection “Rab8 activates while Rab10 inhibits cilia formation in A549 cells”, the authors note that cilia were examined in RPE cells following knockdown of Rab8A and Rab10, but the data are not shown.

2) Several sections of the Results are confusing and should be revised (see Minor Comments 1-5 for details).

3) While the authors have tested the functional interdependencies among LRRK2, Rab8A, Rab10, and RILPL1, the bigger picture of how these proteins interact to control ciliogenesis remains unclear with respect to where Rab8A fits in. Could the authors integrate Rab8A into a pathway model like that in Figure 6D? It seems that one possible model is RILPL1 --| Rab8A --| Rab10 --| ciliogenesis, with LRRK2 promoting Rab10 activity, but this diverges a bit from Figure 6D and perhaps there are alternatives that fit better or otherwise merit consideration. Lastly, given the evidence that Rab10 can inhibit ciliogenesis in the absence of RILPL1, are there Rab10 effectors that the authors view as good candidates for mediating Rab10's inhibition of ciliogenesis?

Minor Comments:

1) The authors state, "addition of exogenous Rab10 suppresses cilia formation in cells lacking Rab8A or Rab10. Moreover, Rab8A enhances cilia formation in cells lacking Rab10; thus this represents an independent pathway for ciliogenesis activation". First, the effect of exogenous Rab10 expression on ciliogenesis in Rab8A-deficient cells does not appear to have been examined here. Second, the data indicate that Rab8A does not enhance cilium formation in cells lacking Rab10; rather, Figure 2B shows that cells lacking Rab10 bypass Rab8A's normal role in promoting ciliogenesis – itself a striking finding. Finally, this section concludes by discussing evidence from Sato et al. that Rab8A/B double-mutant mice show normal primary cilia. This fact seems to confuse or contradict the present data; it may be helpful instead to discuss the findings of Sato et al., 2014, in the Discussion.

2) The conclusions that can be drawn from the use of phospho-site T->A and T->E mutants are also not clear. As the authors note, the Rab10A T73A appears to be non-functional and should not be used as a phosphorylation-blocking allele. Given that Rab8A T72A and T72E mutants both fail to rescue ciliogenesis defects in Rab8A knockout cells and interact poorly with RILPL1 (while phosphorylated Rab8A robustly binds RILPL1), it is likely these mutants are non-functional and do not specifically block or mimic phosphorylation. Thus, the statement in the Abstract that "Rab8A phosphorylation blocks its ability to promote ciliogenesis" should be revised. Also, is the more potent inhibition of ciliogenesis by the Rab10 T73E mutant relative to wildtype Rab10 statistically significant?

3) The order of data presented in the text does not closely follow the order of data in the figures, especially for the first 2 figures and Figure 3—figure supplement 1. Adjusting the order of the figure panels would make the paper easier to follow. Also, in the first paragraph of the subsection “RILPL1 regulates pRab10 localization”, the reference to Figure 3—figure – supplement 1 appears to instead refer to Figure 5—figure – supplement 1. In the first paragraph of the subsection “RILPL1 regulates pRab10 localization”, the reference to Figure 4B appears to refer instead to Figure 4A. Lastly, what is being shown in Figure 2A with respect to GFP-Rab8 and GFP-Rab10?

4) An additional area of confusion is in the section titled, "RILPL1 suppresses cilia formation via centrioles". It seems from the data in Figure 5F-H that, contrary to the section title, RILPL1 suppresses cilia formation independently of effects on centriole spacing. Additionally, while the effects on ciliogenesis of RILPL1 expression in RILPL1 KO cells are striking, it would be informative to assess whether a similar inhibition of ciliogenesis occurs when RILPL1 is over-expressed in wildtype cells.

5) In Figure 4, the authors analyze the effects of RILPL1 fragments on pRab10 localization. While the reported effects are interesting, the suggestion "that the RILPL1 N-terminus interacts with pRab10 in pericentriolar membranes" should perhaps be tempered given that a prior report from these groups showed that it is the C-terminal RH domain in RILPL1 that mediates interaction with Rab8A (PMID: 29125462), and the same may be true for Rab10 (even considering the data shown).

6) The results from LRRK2-G2019S iPS cells in Figure 7C-E could be attributed to differences in ciliogenesis associated with the process of deriving the mutant versus wildtype lines. This potential concern could be alleviated by examining whether MLi2 treatment restores normal levels of cilium formation to the G2019S cells.

7) The authors state that "Yoshimura et al., 2007, were the first to show that Rab8A, but not Rab8B, is the only Rab GTPase to localize to primary cilia in hTert-RPE1 cells". This statement may be misleading to some readers given subsequent evidence that GFP-Rab23 localizes to RPE1 cilia (PMID: 26136363), as does Rab8B (PMID: 21273506). Additionally, while Rab8 and Rabin8 promote ciliogenesis, this effect appears to be only partly through the BBSome and may instead be primarily related to delivery of membrane vesicles to the growing cilium (e.g. PMID: 25812525).

*Reviewer #3:*

This paper evaluates the role of the Rab8 and Rab10 GTPases as well as the Rab effector RILPL1 in the regulation of ciliogenesis by the LRRK2 kinase. It then goes on to explore the role of changes in ciliogenesis in mediating the effects of LRRK2 in Parkinson’s disease. In general the studies are well designed and well executed and the results are by in large convincing. While the paper probably opens as many questions as it closes, it represents an important advance in understanding one molecular mechanism underlying a genetic cause of Parkinsons and should have a broad audience. Below I have outlined a number of points that should be explored more thoroughly or, in one case, deleted for clarity.

1) In Figure 1D and E what is the level of expression of Rab8 and 10 and their mutant alleles relative to the endogenous level? It looks very high.

2) The relationship of Rab8 to Rab10 has not been explored in depth. Does the loss or overexpression of one affect the localization of the other at the cilia? Does the loss or overexpression of one affect the activation of the other? They are closely related and can both bind RILPL, how does each affect the interaction of the other with this common effector? They interpret their data to indicate that the effects of each on ciliogenesis are totally independent of the other, but I am not so sure.

3) In Figure 2B, C and D a WT control is needed.

4) The text refers to cells in which a component is knocked out by CRISPR as "depleted". It would be more accurate to say that the component is deleted. Depletion suggests a partial loss.

5) In Figure 2E they should also test the membrane association of the Rab8 mutants. Directly testing for isoprenylation of the various Rab8 and 10 alleles would address the issue more clearly.

6) Figure 3—figure supplement 1 looks at the effect of phosphorylation on the interaction of Rab8 with RILPL, however the effects on ciliogenesis are predominantly mediated by Rab10. They should also explore the effects of phosphorylation of Rab10 on its interaction with RILPL.

7) The discussion of the effects of RILPL on centriolar distance is very confusing and appears to be of no obvious relevance to this story. In Figure 5G and H it looks like there is an effect, but in 5I it looks like there isn't a significant effect. This is confusing. I would suggest deletion of this section of the paper as it distracts from the main line of reasoning.

---

## [Author Response]

Major concerns:1) It is not firmly established that Rab8 and Rab10 play important roles in ciliogenesis in cell types other than the A549 cell line (see reviewer 2, major comment 1), which is not commonly used for investigating ciliogenesis. The impact of the work could be better evaluated by repeating a limited number of key experiments in RPE cells or wildtype MEFs (e.g. to show that Rab10/RILPL1 can inhibit ciliogenesis in these cells). This seems to have been addressed with experiments done using RPE cells (subsection “Rab8 activates while Rab10 inhibits cilia formation in A549 cells”, first paragraph), but the data is not shown. At a minimum, this data should be shown.

We now include the requested data – the effects are indeed seen in well ciliated cells such as RPEs and MEFs (new Figure 1C, Figure 5D, Figure 6C right).

2) Neither the Rab8 or Rab10 T-to-E/A mutant proteins are functional and it seems likely that the mutant Rabs are non-functional independent of their presumed effects on phosphorylation, and this point is not adequately explored or clearly conveyed. It is shown that Rab10 T73A is cytosolic, explaining why it is non-functional. Why are the other rab8 and Rab10 mutants non-functional? Are they isoprenylated? If the mutants do not selectively affect phosphorylation (e.g. they alter isoprenylation), their use seems tangential to the main point of the paper.

As requested, we omitted our experiments using the mutant proteins since the results are not properly interpretable.

We have consolidated the mutant characterization data to a single place in the manuscript and include information on their binding or lack thereof to RILPL1 (Figure 2—figure supplement 1) and now add data on their in vitro prenylation (Figure 2C) and more fractionation (Figure 2B). This data is important to include for the field, as many studying phosphorylation rely on phosphomimetic mutants or non-phosphorylatable proteins without studying them in greater detail. We thank the reviewers for guiding us to this clearer means of presentation.

3) The effect of the Rab10 LRRK2 phosphorylation site mutations (T73A, E) on its association with RILP1 should be determined, similar to the analysis of Rab8 mutants and RILP1 (Figure 3—figure supplement 1). This information is important for interpreting the phenotypes of the mutant proteins.

We now include this requested data, which belongs here, as the reviewers noted (Figure 2—figure supplement 1B).

4) Please note that each of the reviewers were confused by certain aspects of the presentation (see the reviews). I encourage you to consider revising the presentation to address these sources of confusion.

We have re-organized the manuscript in response to each of the reviewer’s comments.

Separate reviews (please respond to each point):

Reviewer #1:

This study addresses the role of LRRK, a protein kinase implicated in Parkinson's disease, in ciliogenesis. Mutations in LRRK that cause PD activate its protein kinase activity, but how this leads to PD is unknown. The study follows an earlier publication that reported that Rab proteins are the principal substrates for LRRK, including Rab8 and Rab10, the focus of this study. Rab8 has previously been implicated in ciliogenesis, and the authors present evidence that phosphorylation of Rab8 by LRRK inhibits its function in ciliogenesis. In contrast, the authors find that depletion of Rab10, another Rab GTPase implicated in ciliogenesis, potentiates ciliogenesis and that phosphorylation of Rab10 by LRRK potentiates its inhibitory effect on ciliogenesis via enhanced interaction with RILPL1, a Rab10 effector of unknown function. It is shown that depletion of RILPL1 potentiates ciliogenesis, similar to depletion of Rab10. Finally, it is shown that PD mutations in LRRK cause a decrease in the proportion of ciliated cells and deficient SSH signaling in striatal cholinergic neurons of the mouse brain. The specific functions of Rab8 or Rab10 in ciliogenesis or PD are largely unknown. This is a technically sound, well controlled study, that clearly supports roles for Rab8 and Rab10 in ciliogenesis, though if these GTPases play direct, specific roles in ciliogenesis remains an open question. In my opinion, the major value of this study lies in the identification of functional consequences for LRRK-mediated phosphorylation of Rab8 and Rab10 in ciliogenesis, a finding that puts a spotlight on ciliogenesis for future studies of PD etiology.Phenotypic analyses of Rab8a/b knockout mice by Sato et al., 2014, showed essentially no effect on cilia number and little evidence of ciliopathies that would be expected by the loss of a key factor in ciliogenesis. Moreover, they reported that knockout of Rab10 resulted in a loss of cilia, the opposite reported here. How do the authors reconcile these published observations with their results? If I understand correctly, the authors propose that Rab8 "activates" ciliogenesis, rather than provides an essential role, but it is unclear to me what "activate" means.

We removed the word activate and replaced it with increases.

Sato et al. reported no change in cilia number of length upon Rab8A/Rab8B knockout, and additional Rab10 siRNA decreases ciliation 15%. These authors never show the effect of Rab10 KO alone in their wild type background – it is always in the background of double knockout. It is possible that the loss of Rab8B or the double knockdown of Rab10 and Rab13 explains the differences from what we observe and their findings and we have tried to clarify the text in the Discussion.

It is shown that loss of RILPL1 potentiates ciliogenesis, similar to loss of Rab10. It is also shown that Rab10 does not require RILPL1 for its effects on ciliogenesis ("This suggests that Rab8A and Rab10 do not require RILPL1 for their effects on ciliogenesis…"). So why are Rab10 and RILPL1 proposed ("LRRK2 kinase acts via a pRab10-RILPL1 pathway to inhibit ciliogenesis") to function on the same ciliogenesis pathway? It would be helpful if the authors clarified this conclusion.

We have clarified the text – “Figure 6D compares two scenarios by which Rab10 and RILPL1 interfere with cilia formation. At endogenous levels of expression, we propose that LRRK2 generates an inhibitory complex that contains pRab10 and RILPL1. RILPL1 shows enhanced binding to p-Rab10, and our data show that LRRK2 inhibition requires both of these proteins to yield a ciliation defect. In addition, in the absence of LRRK2, upon overexpression of RILPL1, we also detect inhibition of cilia formation; this inhibition requires the presence of Rab10, and may reflect a similar inhibitor complex that requires overexpression to generate in the absence of Rab10 phosphorylation. On the other hand, overexpressed Rab10 can also inhibit cilia formation in a RILPL1-independent manner, that may involve RILPL2 or other proteins. Further work is needed to explore this possibility.”

Additional data files and statistical comments:Rigour and extent of the statistical information are satisfactory.

Reviewer #2:

[…] The negative regulation of ciliogenesis by Rab10/RILPL1 is an interesting finding that, if generalizable to other cell types/tissues, represents a novel insight into ciliogenesis that motivates future analysis of the underlying mechanism. Additionally, the findings that LRRK2 hyper-activity inhibits ciliogenesis and blunts Hedgehog signaling provide further evidence that ciliary defects may contribute to Parkinson's Disease. Lastly, the observation that Rab10 knockdown suppresses ciliogenesis defects seen in MEFs with elevated LRRK2 activity provides an important connection between the two aforementioned findings. As detailed below, the study could be strengthened by further examining Rab10's role in ciliogenesis in cell lines that are more commonly used to study cilia. Additionally, some data are presented in a confusing manner and do not support some of the conclusions reached. These unclear sections significantly obscure the key results and should be revised before publication in eLife.1) Many of the experiments are done in A549 cells, but A549 cells ciliate at a relatively low rate (~25%) and are not commonly used to study ciliogenesis. Thus, it is possible that the inhibitory effects on ciliogenesis of Rab10/RILPL1 may be a distinct feature of this cell type rather than indicative of a more general role for these proteins in regulating cilium formation (especially as Sato et al., 2014, reported Rab10 to be instead a positive regulator of ciliogenesis that acts redundantly with Rab8A and Rab8B). This concern is partially mitigated by the data from LRRK2-R1441G MEFs shown in Figure 6, but it would nonetheless be helpful to repeat a limited number of key experiments in RPE cells or wildtype MEFs (e.g. to show that Rab10/RILPL1 can inhibit ciliogenesis in these cells). In the first paragraph of the subsection “Rab8 activates while Rab10 inhibits cilia formation in A549 cells”, the authors note that cilia were examined in RPE cells following knockdown of Rab8A and Rab10, but the data are not shown.

We agree and include now data in other cell types that were only omitted for space reasons. Also, note that in professional ciliated cells like RPE, one cannot as easily detect a stimulation of ciliation since most cells start ciliated. As above, we include the data – the effects are indeed seen in well ciliated cells such as RPEs and wt MEFs (new Figure 1C, Figure 5D, Figure 6C right).

2) Several sections of the Results are confusing and should be revised (see Minor Comments 1-5 for details).3) While the authors have tested the functional interdependencies among LRRK2, Rab8A, Rab10, and RILPL1, the bigger picture of how these proteins interact to control ciliogenesis remains unclear with respect to where Rab8A fits in. Could the authors integrate Rab8A into a pathway model like that in Figure 6D? It seems that one possible model is RILPL1 --| Rab8A --| Rab10 --| ciliogenesis, with LRRK2 promoting Rab10 activity, but this diverges a bit from Figure 6D and perhaps there are alternatives that fit better or otherwise merit consideration. Lastly, given the evidence that Rab10 can inhibit ciliogenesis in the absence of RILPL1, are there Rab10 effectors that the authors view as good candidates for mediating Rab10's inhibition of ciliogenesis?

We added a new experiment (Figure 1G) that shows that Rab8A can still increase ciliation in the absence of Rab10. We thus cannot yet place it in our pathway. As for other effectors, RILPL2 is obvious (Steger et al., 2017) but this is beyond the scope of this paper.

Minor Comments:1) The authors state, "addition of exogenous Rab10 suppresses cilia formation in cells lacking Rab8A or Rab10. Moreover, Rab8A enhances cilia formation in cells lacking Rab10; thus this represents an independent pathway for ciliogenesis activation". First, the effect of exogenous Rab10 expression on ciliogenesis in Rab8A-deficient cells does not appear to have been examined here.

Thank you for catching this error! We corrected the text as follows: “Thus, Rab10 is a dominant suppressor of cilia formation. Moreover, endogenous Rab8A enhances cilia formation in cells lacking Rab10 (from ~25% ciliation to ~50% ciliation); in addition, exogenous Rab8A even further increases ciliation in the cells lacking Rab10 protein (Figure 1F). Thus, Rab8A drives a Rab10 independent pathway for cilia formation.”

Second, the data indicate that Rab8A does not enhance cilium formation in cells lacking Rab10; rather, Figure 2B shows that cells lacking Rab10 bypass Rab8A's normal role in promoting ciliogenesis – itself a striking finding.

This is now Figure 1F. It shows that loss of Rab10 in a Rab8A KO gives enhanced cilia formation without either Rab—cilia formation is not always Rab8A dependent. Rab8A pathway can act on top of this to enhance cilia formation (new Figure 1G).

Finally, this section concludes by discussing evidence from Sato et al. that Rab8A/B double-mutant mice show normal primary cilia. This fact seems to confuse or contradict the present data; it may be helpful instead to discuss the findings of Sato et al., 2014, in the Discussion.

Good suggestion, we moved this to the Discussion.

2) The conclusions that can be drawn from the use of phospho-site T->A and T->E mutants are also not clear. As the authors note, the Rab10A T73A appears to be non-functional and should not be used as a phosphorylation-blocking allele. Given that Rab8A T72A and T72E mutants both fail to rescue ciliogenesis defects in Rab8A knockout cells and interact poorly with RILPL1 (while phosphorylated Rab8A robustly binds RILPL1), it is likely these mutants are non-functional and do not specifically block or mimic phosphorylation. Thus, the statement in the Abstract that "Rab8A phosphorylation blocks its ability to promote ciliogenesis" should be revised.

Point taken.

Also, is the more potent inhibition of ciliogenesis by the Rab10 T73E mutant relative to wildtype Rab10 statistically significant?

Yes but we removed the data from the paper because the mutants are nonfunctional.

3) The order of data presented in the text does not closely follow the order of data in the figures, especially for the first 2 figures and Figure 3—figure supplement 1. Adjusting the order of the figure panels would make the paper easier to follow. Also, in the first paragraph of the subsection “RILPL1 regulates pRab10 localization”, the reference to Figure 3—figure supplement 1 appears to instead refer to Figure 5—figure supplement 1. In the first paragraph of the subsection “RILPL1 regulates pRab10 localization”, the reference to Figure 4B appears to refer instead to Figure 4A. Lastly, what is being shown in Figure 2A with respect to GFP-Rab8 and GFP-Rab10?

We have fixed this problem as requested.

4) An additional area of confusion is in the section titled, "RILPL1 suppresses cilia formation via centrioles". It seems from the data in Figure 5F-H that, contrary to the section title, RILPL1 suppresses cilia formation independently of effects on centriole spacing. Additionally, while the effects on ciliogenesis of RILPL1 expression in RILPL1 KO cells are striking, it would be informative to assess whether a similar inhibition of ciliogenesis occurs when RILPL1 is over-expressed in wildtype cells.

We have added the requested data and changed the heading and moved the centrosome cohesion to Figure 5—figure supplement 1 and omitted it from the Discussion.

5) In Figure 4, the authors analyze the effects of RILPL1 fragments on pRab10 localization. While the reported effects are interesting, the suggestion "that the RILPL1 N-terminus interacts with pRab10 in pericentriolar membranes" should perhaps be tempered given that a prior report from these groups showed that it is the C-terminal RH domain in RILPL1 that mediates interaction with Rab8A (PMID: 29125462), and the same may be true for Rab10 (even considering the data shown).

Unfortunately the mutants reported in those experiments were later found to be mislocalized to the nucleus so may have lost binding capacity for the wrong reason.

6) The results from LRRK2-G2019S iPS cells in Figure 7C-E could be attributed to differences in ciliogenesis associated with the process of deriving the mutant versus wildtype lines. This potential concern could be alleviated by examining whether MLi2 treatment restores normal levels of cilium formation to the G2019S cells.

We agree and as requested we have now added the MLi2 control, which confirms our earlier conclusions (new Figure 7C).

7) The authors state that "Yoshimura et al., 2007, were the first to show that Rab8A, but not Rab8B, is the only Rab GTPase to localize to primary cilia in hTert-RPE1 cells". This statement may be misleading to some readers given subsequent evidence that GFP-Rab23 localizes to RPE1 cilia (PMID: 26136363), as does Rab8B (PMID: 21273506). Additionally, while Rab8 and Rabin8 promote ciliogenesis, this effect appears to be only partly through the BBSome and may instead be primarily related to delivery of membrane vesicles to the growing cilium (e.g. PMID: 25812525).

Text revised as suggested.

Reviewer #3:

This paper evaluates the role of the Rab8 and Rab10 GTPases as well as the Rab effector RILPL1 in the regulation of ciliogenesis by the LRRK2 kinase. It then goes on to explore the role of changes in ciliogenesis in mediating the effects of LRRK2 in Parkinsons disease. In general the studies are well designed and well executed and the results are by in large convincing. While the paper probably opens as many questions as it closes, it represents an important advance in understanding one molecular mechanism underlying a genetic cause of Parkinsons and should have a broad audience. Below I have outlined a number of points that should be explored more thoroughly or, in one case, deleted for clarity.1) In Figure 1D and E what is the level of expression of Rab8 and 10 and their mutant alleles relative to the endogenous level? It looks very high.

We omitted the mutant alleles because the proteins are nonfunctional; in A549 rescue experiments, the Rabs are expressed 4X (Rab10) or 10 fold (Rab8A) over endogenous (now added to the subsection “Rab8A increases while Rab10 inhibits cilia formation in A549 cells”, first paragraph).

2) The relationship of Rab8 to Rab10 has not been explored in depth. Does the loss or overexpression of one affect the localization of the other at the cilia? Does the loss or overexpression of one affect the activation of the other? They are closely related and can both bind RILPL, how does each affect the interaction of the other with this common effector? They interpret their data to indicate that the effects of each on ciliogenesis are totally independent of the other, but I am not so sure.

Rab8 goes to the cilium fine in Rab10KO cells. We cannot say if Rab10 is localized at the base of the cilium in Rab8A KO cells as they are only poorly ciliated. The levels of each protein are unchanged upon KO of the other Rab (Figure 1—figure supplement 1). We added this in the last paragraph of the subsection “Rab8A increases while Rab10 inhibits cilia formation in A549 cells”.

3) In Figure 2B, C and D a WT control is needed.

Included as requested in what is now Figure 1G.

4) The text refers to cells in which a component is knocked out by CRISPR as "depleted". It would be more accurate to say that the component is deleted. Depletion suggests a partial loss.

Fixed as requested throughout.

5) In Figure 2E they should also test the membrane association of the Rab8 mutants. Directly testing for isoprenylation of the various Rab8 and 10 alleles would address the issue more clearly.

Done as requested (new Figure 2B, C; mutants omitted).

6) Figure 3—figure supplement 1 looks at the effect of phosphorylation on the interaction of Rab8 with RILPL, however the effects on ciliogenesis are predominantly mediated by Rab10. They should also explore the effects of phosphorylation of Rab10 on its interaction with RILPL.

We have now added this as requested (Figure 2—figure supplement 1B).

7) The discussion of the effects of RILPL on centriolar distance is very confusing and appears to be of no obvious relevance to this story. In Figure 5G and H it looks like there is an effect, but in 5I it looks like there isn't a significant effect. This is confusing. I would suggest deletion of this section of the paper as it distracts from the main line of reasoning.

The group of Sabine Hilfiker published that she believes LRRK2 interferes with centrosomal cohesion, which is why we have included this data. We moved it to Figure 5—figure supplement 1 so that it does not detract from the current story, and we no longer discuss it in the Discussion.

We thank the reviewers for their very thoughtful and careful evaluations!